# SPARTA: Scalable and Principled Benchmark of Tree-Structured Multi-hop QA over Text and Tables

**Sungho Park,    Jueun Kim,    Wook-Shin Han***
POSTECH, Pohang, Republic of Korea
`{shpark,jekim,wshan}@dblab.postech.ac.kr`

## Abstract

Real-world Table–Text question answering (QA) tasks require models that can reason across long text and source tables, traversing multiple hops and executing complex operations such as aggregation. Yet existing benchmarks are small, manually curated—and therefore error-prone—and contain shallow questions that seldom demand more than two hops or invoke aggregations, grouping, or other advanced analytical operations expressible in natural-language queries. We present SPARTA, an end-to-end construction framework that automatically generates large-scale Table–Text QA benchmarks with lightweight human validation, requiring only one quarter of the annotation time of HybridQA. The framework first constructs a reference fact database by enriching each source table with grounding tables whose tuples are atomic facts automatically extracted from the accompanying unstructured passages, then synthesizes nested queries whose number of nested predicates matches the desired hop count. To ensure that every SQL statement is executable and that its verbalization yields a fluent, human-sounding question, we propose two novel techniques: provenance-based refinement, which rewrites any syntactically valid query that returns a non-empty result, and realistic-structure enforcement, which confines generation to post-order traversals of the query graph. The resulting pipeline produces thousands of high-fidelity question–answer pairs covering aggregations, grouping, and deep multi-hop reasoning across text and tables. On SPARTA, state-of-the-art models that reach over 70 F1 on HybridQA or over 50 F1 on OTT-QA drop by more than 30 F1 points, exposing fundamental weaknesses in current cross-modal reasoning. Our benchmark, construction code, and baseline models are available at github.com/pshlego/SPARTA.

## 1 Introduction

Table–Text QA has emerged as a fundamental challenge in building robust question answering (QA) systems capable of operating across heterogeneous data modalities (i.e., text and tables) Chen et al. (2020a;b; 2021); Zhao et al. (2022); Zhu et al. (2021). Such a task is particularly evident in scenarios where textual descriptions and table entries originate from one or more sources (e.g., textual information and tables in multiple Wikipedia pages) and must be jointly analyzed to arrive at the correct answer. While a single Wikipedia page often contains both text and tables, it is not unusual for relevant information to span multiple pages or documents, necessitating cross-document retrieval and the effective integration of disparate information.

A significant limitation of existing Table-Text QA benchmarks is that human annotators manually construct them Chen et al. (2020a;b; 2021); Zhao et al. (2022); Zhu et al. (2021), resulting in fundamentally flawed benchmark designs that hinder comprehensive system evaluation.

**(1) Limited question types and shallow reasoning.** Existing Table-Text QA benchmarks, constrained by manual annotation complexities, feature a restricted range of shallow questions. These typically require only direct information extraction (such as pinpointing a fact within a single textual passage or locating a specific entry in a table). Even for questions that go beyond this simple extraction, the

---

*Corresponding author.

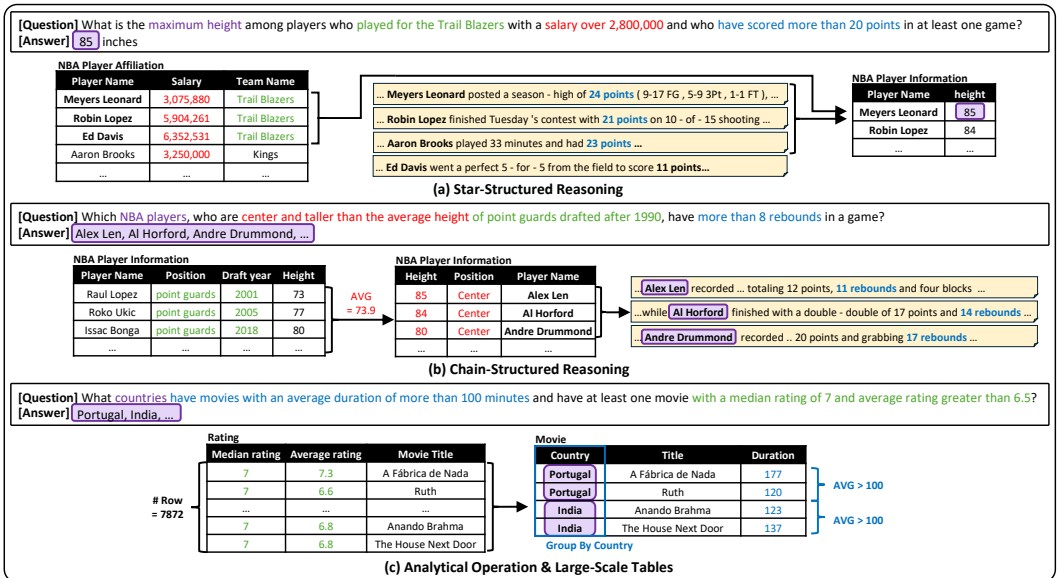

Figure 1: Representative examples of our SPARTA benchmark (see Appendix M for more examples).

| Benchmark | Table Size | | Question Generation | Grouping/Having | Query Shape Supported | | Multi-hop Reasoning | | Annotation Error Rate (over 100 sampled queries) |
|---|---|---|---|---|---|---|---|---|---|
| | #Col | #Row | | | Chain (>3-Hop) | Star | Cross-modal | Uni-modal | |
| TAT-QA Zhu et al. (2021) | 4.0 | 9.4 | Manual | ✗ | ✗ | ✗ | ✓ | ✗ | 30% |
| FinQA Chen et al. (2021) | – | 6.4 | Manual | ✗ | ✗ | ✗ | ✓ | ✗ | 27% |
| MULTIHIERTT Zhao et al. (2022) | 5.0 | 10.8 | Manual | ✗ | ✗ | ✗ | ✓ | ✓ | 26% |
| HybridQA Chen et al. (2020b) | 4.4 | 15.7 | Manual | ✗ | ✗ | ✗ | ✓ | ✗ | 21% |
| OTT-QA Chen et al. (2020a) | 4.4 | 15.7 | Manual | ✗ | ✗ | ✗ | ✓ | ✗ | 21% |
| **SPARTA (NBA)** | **12.2** | **3,280.5** | | | | | | | |
| **SPARTA (Movie)** | **4.7** | **10,054.0** | **Auto (LLM) w/ Lightweight Human Validation** | **✓** | **✓** | **✓** | **✓** | **✓** | **0%** |
| **SPARTA (Medical)** | **6.7** | **200.0** | | | | | | | |

Table 1: Comparison of Table–Text QA benchmarks (see Appendix A for detailed annotation audit results).

reasoning depth remains shallow, seldom demanding more than two hops or involving advanced analytical operations like aggregation or grouping. This is despite such operations being common in real-world natural language queries yet underrepresented in benchmarks. This deficiency hinders the thorough evaluation of a system's deep, multi-step inference capabilities. Furthermore, current multi-hop questions usually follow simplistic linear chains, rather than the expressive, tree-structured reasoning (e.g., multi-branch paths, longer chains, or uni-modality hops) crucial for assessing systems on complex inference tasks, as exemplified in Figure 1.

**(2) Annotation noise.** Our quality audit uncovers numerous annotation errors that undermine the reliability of the benchmark. Re-inspecting 100 randomly sampled dev examples from HYBRIDQA, we find that $21\%$ contain at least one error, which we classify into three categories: (1) *Redundant modality* ($52.4\%$): table and passage encode the same fact, yet the instance is tagged as a cross-modal question even though a single modality suffices; (2) *Incomplete answer set* ($23.8\%$): several answers are correct but only one is recorded, distorting recall; (3) *Incorrect or unanswerable* ($23.8\%$): the labelled answer is wrong or cannot be derived from the provided data, revealing a lapse in quality control. Our audits on other benchmarks reveal similar error patterns (see Appendix A).

**(3) Reliance on single, small-scale web tables.** Current benchmarks almost exclusively draw on compact web tables—typically scraped from Wikipedia or corporate reports—thereby providing only toy-scale scenarios. As Table 1 shows, tasks either involve a single table or, when multiple tables are present, the mean table cardinality hovers around 15 rows, far short of the thousands of rows found in real-world databases. This simplification is largely pragmatic: reasoning over larger tables dramatically increases annotator effort and error rates Chen et al. (2020b). Consequently, existing benchmarks cannot meaningfully evaluate QA systems in realistic, high-complexity settings that demand reasoning over large, heterogeneous relational data.

SPARTA unifies all evidence—structured and unstructured—inside a single relational store called the *reference fact database*. Each original relation (e.g., a web table or a financial ledger) remains intact as a *source table*. *Grounding tables*, which store atomic propositions as tuples for SQL-addressable access, are populated using two complementary methods (detailed in Section 3.2): (1) utilizing validated corpora such as ROTOWIRE Wu et al. (2022); and (2) employing a table-to-text strategy that generates atomic facts directly from structured data. With textual facts now addressable via SQL, queries over this combined store freely mix modalities; no pointer to the original span is needed as answers are returned directly by query execution.

**Stage 1 – Reference fact database construction.** Source and grounding tables are merged into the reference fact database, making all facts uniformly queryable.

**Stage 2 – Query generation.** A large language model (LLM) receives the schema and sample rows and emits SQL whose *number of nested predicates matches a target hop count*. Note that SPARTA synthesizes queries that instantiate the four representative nesting patterns—Types N, A, J, and JA—outlined in Appendix B. Two safeguards ensure that only realistic, executable statements survive: (1) *Provenance-based refinement* loops provenance feedback—unmatched joins or overly selective predicates—back to the LLM until the query returns a non-empty result. (2) *Realistic-structure enforcement* confines generation to post-order traversals of query graph, yielding human-like join orders and enabling early pruning of infeasible subqueries.

**Stage 3 – Question verbalisation.** Each validated query is paired with its execution result, then a second LLM rewrites the SQL into a fluent natural-language question, producing high-fidelity pair ⟨question, answer⟩ that span aggregation, grouping, and deep multi-hop joins across large tables. Here, the final correctness—i.e., the validity of the question–answer pair—is checked via lightweight human verification; unlike HybridQA, our pipeline does not require re-performing full multi-hop reasoning, thereby keeping audit costs low (see Section 3.4).

This SQL-centric pipeline yields a large, diverse, and rigorously validated benchmark that corrects the size, noise, and logical shallowness of previous Table–Text QA resources. On SPARTA, state-of-the-art models that exceed 70 F1 on HybridQA or exceed 50 F1 on OTT-QA drop by more than 30 F1 points, revealing fundamental weaknesses in current cross-modal reasoning and highlighting directions for future research.

## 2 RELATED WORK

**Table-Text QA Benchmark.** Table–Text QA benchmarks evaluate a model's ability to jointly reason over structured tables and unstructured passages. HybridQA Chen et al. (2020b) introduced the task, and OTT-QA Chen et al. (2020a) extended it to open-domain settings, but both suffer from annotation noise, shallow reasoning depth, and a lack of support for advanced analytical operations. Specifically, they do not support GROUP BY or HAVING clauses, and only 1.1% of questions involve aggregation. Their multi-hop reasoning is confined to short, linear chains and fails to capture tree-structured or uni-modal reasoning paths. Other benchmarks—TAT-QA Zhu et al. (2021), FinQA Chen et al. (2021), and MultiHiertt Zhao et al. (2022)—focus narrowly on numerical reasoning in financial contexts rather than multi-hop reasoning, further limiting coverage Zhang et al. (2023). Additionally, all existing Table-Text QA datasets rely on small, manually annotated web tables, which hinders scalability and realism. SPARTA addresses these gaps with an SQL-centric pipeline that constructs a large-scale benchmark of executable, compositional questions over hybrid corpora, offering a principled testbed for multi-hop QA across text and tables.

**Synthetic Benchmark Generation.** Recent synthetic benchmark generation scales QA pairs from pre-existing sources, but most are single-modal: relying on knowledge graphs Sun et al. (2024); Omar et al. (2025); Orogat & El-Roby (2023; 2022) or text corpora Bonifacio et al. (2022); Jeronymo et al. (2023), ignoring cross-modal reasoning. ERBench Oh et al. (2024) uses relational databases, yet its questions are binary or multiple-choice, based on shallow templates excluding analytical operators like GROUP BY, HAVING, and aggregations; it also lacks table-text interplay. Similarly, TDBench Kim et al. (2026) leverages temporal databases to automate time-sensitive QA, but it is confined to temporal reasoning within structured tables. In contrast, SPARTA generates multi-hop questions bridging tables and passages, mirroring complex nested SQL patterns to provide a rigorous cross-modal benchmark for Table–Text QA. Beyond QA, benchmarks in other domains impose

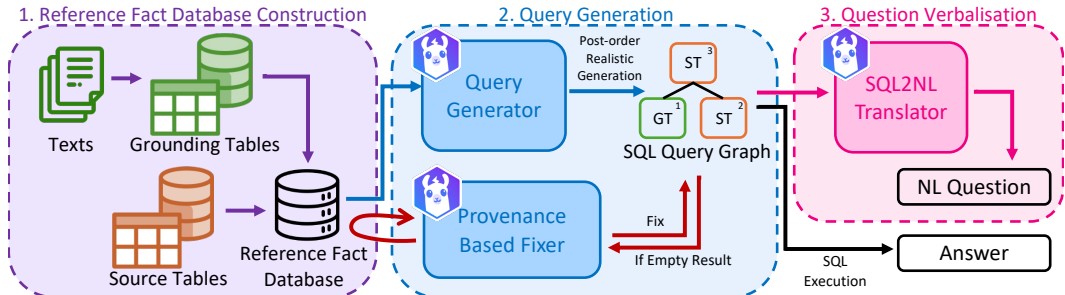

Figure 2: Overview of SPARTA: (1) Reference Fact Database Construction, (2) Query Generation, (3) Question Verbalisation. ST and GT denote a source table and a grounding table, respectively.

domain-specific constraints: database performance benchmarks Nambiar & Poess (2006); Erling et al. (2015) use fixed schemas and templates for reproducible profiling; unlearning benchmarks Maini et al. (2024); Zhong et al. (2024) create forget/retain partitions for selective forgetting; and PEEL Kim (2024) employs template-based generation of NL-Nested SQL pairs to guarantee the executability of structurally complex queries. SPARTA's constraint is fundamentally different: every synthetic example must encode tree-structured multi-hop reasoning grounding semantically sound, executable SQL and natural-language questions, requiring analytical operations and table-text alignment. Our provenance-based refinement and realistic-structure enforcement address this, producing semantically rich, executable queries.

## 3 SPARTA

### 3.1 TABLE–TEXT QA TASK AND BENCHMARK GENERATION

Given a natural-language question $q_{\mathrm{NL}}$, a set of source tables $\mathcal{S}_T = \{T^{(1)}, \ldots, T^{(m)}\}$, and a set of passages $\mathcal{C}_P = \{P^{(1)}, \ldots, P^{(n)}\}$, a QA system $f_\theta$ must return the answer $a = f_\theta(q_{\mathrm{NL}}, \mathcal{S}_T, \mathcal{C}_P)$. Each passage in $\mathcal{C}_P$ is decomposed into atomic facts and stored as tuples in *grounding tables* $\mathcal{G}_T$. Merging these with the original source tables yields a unified *reference fact database* $\mathcal{D}$. An LLM then: (i) generates executable SQL queries on $\mathcal{D}$ that vary in depth (selection, aggregation, nesting, etc.), and (ii) verbalises each query into a fluent natural-language question $q_{\mathrm{NL}}$. The resulting pairs $(q_{\mathrm{NL}}, a)$ constitute a scalable benchmark for Table–Text QA. An overview of the entire pipeline is provided in Figure 2.

### 3.2 REFERENCE FACT DATABASE CONSTRUCTION

We use the ROTOWIRE dataset as part of our reference fact database, whose structured tables are widely used as gold supervision for text-to-table and have been verified by the authors of Wu et al. (2022) for consistency with the accompanying game reports. Each NBA game report in this corpus is decomposed into atomic facts, which are stored as tuples in $\mathcal{G}_T$, guaranteeing perfect alignment between text and relational data. To construct $\mathcal{S}_T$, we integrate six public NBA datasets—covering salaries, awards, draft data, and team histories—sourced from Kaggle and data.world kag (f;a;g;b;e); dat. Shared entity attributes such as PLAYER_NAME and TEAM_NAME are enforced as primary–foreign key pairs, yielding a connected schema in which every tuple from $\mathcal{G}_T$ can be joined to at least one table in $\mathcal{S}_T$. The resulting database contains three grounding tables and six source tables (see Appendix C).

While our construction uses NBA data for illustration, SPARTA is inherently domain-agnostic. From any relational database, one designates a subset of relations as $S_T$ and treats the remaining relations as $\mathcal{G}_T$. Applying table-to-text generation to $\mathcal{G}_T$ yields a companion set of textual passages $\mathcal{C}_P$, forming the reference-fact database $\mathcal{D} = \mathcal{S}_T \cup \mathcal{G}_T$ with no information overlap between the two sets. The query-generation pipeline then applies unchanged, yielding a portable recipe for building large-scale Table–Text QA benchmarks in any domain with relational data. To demonstrate this, we extended our pipeline to two new domains—movies and medical—using Kaggle datasets kag (c;d), with configurations identical to the NBA domain (see Appendix E). For these datasets, we start from

existing structured tables and convert a subset into grounding tables using rule-based templates. This table-to-text transformation is deterministic and template-driven, with templates manually designed and verified to prevent spurious facts or errors.

### 3.3 QUERY GENERATION

For non-nested queries, SPARTA builds the statement clause-by-clause: the LLM emits each clause in canonical SQL order, conditioned on the schema and previously written clauses, and immediately executes the partial query. If the result is empty, the execution outcome is fed back so the LLM can revise the offending clause, ensuring the query remains executable and semantically meaningful at every step.

The next step is to synthesise nested SQL queries that act as faithful logical forms for multi-hop reasoning. A generated query must satisfy two criteria: (i) it should resemble a query that a human analyst would plausibly write, avoiding degenerate template artifacts, and (ii) it must execute over $\mathcal{D}$ without error and return a non-empty result. These guarantees ensure that every $(q_{\mathrm{NL}}, a)$ pair is both natural and answerable.

Template-based generation fills fixed slots with ad-hoc limits or auxiliary predicates to guarantee execution, yet the resulting SQL is often semantically unsound. For instance, `SELECT birthplace FROM nba_player_information WHERE birthplace <> 'Chicago, Illinois' OR birthplace <> 'Dallas, Texas'` runs without error but expresses a vacuous intent ("... not born in Chicago *or* not born in Dallas," matching everyone). Conversely, one-shot LLM prompting produces natural queries, but these frequently yield empty results and show limited diversity (see Table 3). We therefore introduce a dual-stage framework: (i) *realistic-structure enforcement* and (ii) *provenance-based refinement*.

#### 3.3.1 REALISTIC-STRUCTURE ENFORCEMENT

A nested SQL query can be modeled as a *query graph* $G = (V, E)$ where each node $v_i \in V$ corresponds to a distinct query block—namely every `SELECT ... FROM ... WHERE ...` subquery including the outermost statement—while each (directed) edge $e_{ij} \in E$ denotes a *nested predicate* that correlates blocks $Q_i$ and $Q_j$ through a shared attribute reference, thus capturing the dependency structure of the original nested query in graph form (see Appendix B for representative nested query patterns). Based on this representation, we measure query complexity by the number of edges in the query tree, each representing a reasoning hop.

For nested-query generation, SPARTA adopts *Post-Order+Prov* as the default. That is, to preserve realistic structure, we force the LLM to build the query tree in post-order: compose each leaf subquery first, then wrap it with successively higher-level blocks—exactly how analysts craft nested SQL. We choose post-order traversal over alternatives like breadth-first or top-down, because the latter require validating incomplete queries before inner subqueries are constructed. In contrast, post-order ensures that each intermediate block is executable by validating subqueries first and then composing higher-level predicates. In *Post-Order+Prov*, leaf nodes are generated clause-by-clause. For the target question type we pick the relevant clauses (`WHERE`, `GROUP BY`, `ORDER BY`, ...) in canonical order, and let the LLM fill each one using (i) the schema, (ii) earlier clauses, and (iii) partial results. If a clause yields an empty result, we roll back to the last valid subquery, sparing redundant LLM calls. Internal nodes arise by recursively enclosing validated subqueries. At every step the LLM selects a child query, picks a joinable table, and emits a connecting predicate (`AND`/`OR`, etc.). Empty outputs trigger provenance-guided repair (§3.3.2); otherwise the predicate is kept. The loop iterates until the query graph grows to the specified target size.

#### 3.3.2 PROVENANCE-BASED REFINEMENT

The LLM builds the query graph in post-order—validated leaves first, then one outer predicate at a time. If an evolving query returns no rows, a provenance-based refinement process is initiated to repair the query. The refinement process leverages "why-not provenance," a database technique used to identify which predicates in a query are responsible for filtering out expected tuples Bidoit et al. (2014); Chapman & Jagadish (2009); Lee et al. (2017). While traditional why-not provenance

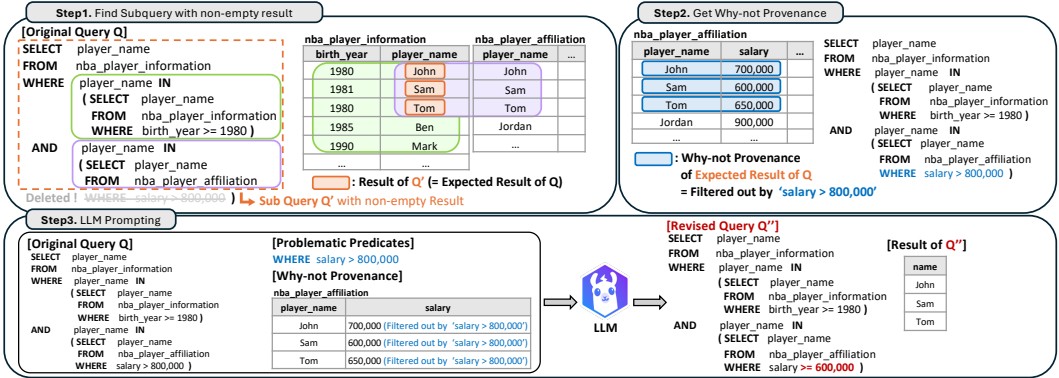

Figure 3: Overview of provenance-based refinement.

often relies on user-provided examples of the missing tuples, our approach dynamically derives the expected tuples from intermediate query results.

The process unfolds in three steps. First, when a query yields an empty result, we peel off predicates in reverse order until the query yields a result. Second, we sample a tuple from this non-empty result set. Finally, we run a why-not provenance tool Dietrich et al. (2022) to identify the blocking predicate and provide this provenance report to the LLM, instructing it to rewrite only the problematic clause.

Ablations are (i) *One-Shot–k*, which inserts all $k$ predicates in a single pass with no checks, and (ii) *Post-Order* (no provenance), which follows the same construction but skips the repair loop. Figure 3 illustrates the overall process of provenance-based refinement. Provenance feedback relaxes the predicate from `salary > 800000` to `salary > 600000`.

## 3.4 QUESTION VERBALISATION

For each executable SQL query $q_{\mathrm{SQL}}$, we generate a corresponding natural-language question $q_{\mathrm{NL}}$ using AST-ICL Al Lawati et al. (2025), a SOTA LLM-based SQL-to-text model. We adopted the LLM-based model over template-based methods, which are limited by rigidity and reliance on handcrafted templates, as documented in prior work Iyer et al. (2016); Xu et al. (2018). In AST-ICL, the SQL abstract syntax tree is supplied as an in-context exemplar, and the model emits a fluent question $q_{\mathrm{NL}}$ whose semantics align with the query. Executing $q_{\mathrm{SQL}}$ on $\mathcal{D}$ yields the answer $a$, completing the benchmark pair $(q_{\mathrm{NL}}, a)$. Every instance is thus interpretable, executable, and suitable for probing multi-hop reasoning over hybrid (table + text) data.

The verbalized questions were validated and corrected by three CS graduate students with SQL/schema literacy to ensure factuality and meaningfulness. This process is lightweight, requiring substantially less effort than full manual annotation. Specifically, validating 3,300 queries takes about 1,493 minutes of total worker time, whereas HybridQA required roughly 6,600 minutes to create the same number of queries from scratch.

## 4 EXPERIMENTS

### 4.1 EVALUATION SETUP

**Hardware and Software Settings.** We conducted our experiments on a machine with Intel(R) Xeon(R) Gold 6230 CPU @ 2.10GHz and 1.5 TB of RAM running Ubuntu 22.04.4 and 4 RTX A6000 GPUs, with LLM inference managed via the SGLang Zheng et al. (2024) inference engine. We used Llama-3.1-70B-Instruct Dubey et al. (2024) as the LLM.

**Query Generation Methods.** For non-nested query generation, SPARTA's default is *Execution-Guided* generation: the LLM writes each clause in canonical order, executes the partial query, and immediately edits any clause that empties the result. As an ablation we also evaluate (i) *One-Shot*,

which emits the whole query from schema only, and (ii) *Clause*, which builds the query sequentially without execution feedback.

For nested-query generation, SPARTA's default is *Post-Order+Prov*: validated leaves are wrapped one predicate at a time (post-order); each new predicate is executed immediately and, when empty, repaired with provenance feedback. Ablations include (i) *One-Shot–$k$*, which inserts all $k$ predicates in a single pass with no intermediate checks, and (ii) *Post-Order* (no provenance), which follows the same post-order construction without provenance-based repair. We generate 500 non-nested and 600 nested SQL queries per method on the NBA domain (configuration as in Table 10), so that quality and cost can be compared on equal footing.

**Table-Text QA Methods.** To gauge how current state-of-the-art systems break down under SPARTA's deeper hops, larger tables, and advanced analytical operations, we evaluate SOTA Table–Text QA methods, including methods based on prompting LLMs such as ODYSSEY Agarwal et al. (2025) and HProPro Shi et al. (2024). These models have shown strong results on HybridQA, where models reason over provided tables and linked documents. ODYSSEY constructs a graph from the table and linked documents, enabling the LLM to traverse the graph for query answers. HProPro generates and executes program via the LLM to produce query responses. Since existing Table–Text QA methods are not originally designed to support uni-modal hops, we apply minimal extensions to enable such behavior during evaluation on SPARTA. Specifically, for ODYSSEY, we augment the hybrid graph by adding edges between matching cells of columns that share a join relationship. For HProPro, we adapt the prompt format by replacing the input table with a list of relevant tables. For a fully end-to-end scenario in which no oracle is provided, we pair the Table–Text QA methods with HELIOS Park et al. (2025)—the top retriever on OTT-QA—so the model must both retrieve evidence and reason over it. We also run every method with GPT-5 and GPT-3.5-turbo backbones to test LLM sensitivity.

## 4.2 BENCHMARK GENERATION COST AND QUERY NATURALNESS

A scalable benchmark must maximise *useful* queries while minimising LLM calls and wall time. We therefore track seven complementary metrics in Table 2.

Table 2: Cost metrics used for benchmark generation.

| Metric | Definition |
|---|---|
| Success-Q | # of non-nested queries that execute without error and return at least one row. |
| Exec-Err | # of statements that fail at parse or runtime, revealing schema or logic errors. |
| Empty-Q | # of syntactically valid queries that return zero rows because predicates are too restrictive. |
| Duplicate-Q | # of queries whose result duplicates a previously generated query, reducing diversity. |
| Ideal Calls | # of LLM invocations required if every step succeeds on the first attempt (baseline cost). |
| Total Calls | # of actual LLM invocations, i.e., Ideal Calls plus extra calls for provenance-guided fixes or other retries. |
| Wall Time | Total wall-clock time to obtain all successful queries. |

Table 3 summarizes generation overheads for both non-nested and nested SQL. For non-nested queries, *Execution-Guided* is most economical, needing only 1,134 total LLM calls—just 7.2% above the ideal 1,058—and finishing in 2,466s. In contrast, *One-Shot* begins with the lowest ideal budget (500 calls) but produces 60 empty and 1,265 duplicate outputs, inflating usage to 1,830 real calls (266% of ideal) and incurring the highest latency; *Clause* mitigates these failures yet still exceeds its ideal by 24.9%. For nested queries, *Post-Order+Prov* is most cost-effective, completing with 4,722 calls in 26,278s—cutting call volume by 42.8% versus vanilla post-order and by 66.2% versus *One-Shot–$k$*. These results show that disciplined post-order construction combined with provenance-driven repair minimizes redundant generations while ensuring executable, semantically plausible SQL; detailed analysis of generation overheads across varying query graph shapes and sizes is provided in Appendix F.

To assess the realism of the generated SQL queries, we employ a scoring-based evaluation framework combining automatic and human assessments. Each query is rated from 1 (least natural) to 5 (most natural) across three dimensions: `Relevance`, which measures alignment with the genuine curiosity of a typical person; `Specificity & Clarity`, which assesses whether the query expresses a clear and well-scoped information need; and `Overall Naturalness`, which combines the above criteria to decide whether the query is likely to be asked by a real person. For a comprehensive assessment, we conduct an automatic evaluation (*auto-eval*) using ChatGPT-4o OpenAI and an independent human evaluation (*human-eval*) by three external CS graduate students with SQL/schema literacy. As a baseline for comparison, we also evaluate queries generated by template filling with randomly sampled column–value pairs. This dual approach, integrating LLM-based auto-evaluation

Table 3: Generation Cost Comparison of Query Generation Methods.

| Method | Success-Q | Empty-Q | Duplicate-Q | Exec-Err | Ideal Calls | Total Calls | Wall Time (s) |
|---|---|---|---|---|---|---|---|
| **Non-nested Query Generation** | | | | | | | |
| *One-Shot* | 500 | 60 | 1265 | 5 | 500 | 1830 | 4256.96 |
| *Clause* | 500 | 51 | 78 | 0 | 1053 | 1316 | 3218.83 |
| *Execution-Guided* | 500 | 0 | 27 | 0 | 1058 | 1134 | 2466.47 |
| **Nested Query Generation** | | | | | | | |
| *One-Shot–k* | 600 | 0 | 0 | 0 | 2664 | 13962 | 115316.67 |
| *Post-Order* (no provenance) | 600 | 0 | 0 | 0 | 3104 | 8253 | 38867.40 |
| *Post-Order+Prov* | 600 | 0 | 0 | 0 | 3074 | 4722 | 26277.87 |

Figure 4: Comparison of Query Naturalness for Different Generation Methods.

with human judgment, yields a robust, multi-perspective measure of how convincingly the generated queries mirror real user intent.

Figure 4 reports the naturalness scores of queries generated by different methods, evaluated across three criteria. Among the non-nested query generation methods, Execution-Guided Generation achieved the highest scores consistently across both automatic and human evaluations. Specifically, in terms of overall naturalness, it outperformed Clause-by-Clause, One-shot, and Template-based generation by 1.3%, 11.4%, and 37.5%, respectively, in auto-eval; and by 6.0% and 36.7% over One-shot and Template-based methods in human-eval. For nested query generation, Post-order Generation with Execution Guidance achieved the top scores across all three metrics. Compared to Post-order, One-shot Nested, and Template-based generation, it yielded auto-eval improvements of 1.7%, 8.1%, and 123.2%, and human-eval gains of 2.1%, 12.5%, and 117.8%, respectively. These results confirm that LLM-based generation strategies—especially those leveraging clause-wise generation and post-order traversal—are significantly more effective at producing realistic and fluent SQL queries than template-based approaches.

### 4.3 TABLE-TEXT QA EVALUATION RESULTS

Table 4 and Table 5 report the Table–Text QA performance of representative methods across eight benchmarks, revealing the increased difficulty posed by SPARTA. We evaluate SPARTA under two configurations: (1) SPARTA (Oracle), where models are given ground-truth tables and linked passages; and (2) SPARTA (Retrieval), where models must retrieve relevant content from the entire corpus. On SPARTA (Oracle), ODYSSEY with GPT-5 achieves an average F1 score of 35.6% across all domains, representing a sharp 33.9-point drop compared to its performance on HybridQA (69.5%). Similarly, HProPro with GPT-5 achieves an average F1 score of 40.4%, a 30.1-point drop from its HybridQA performance (70.5%). These results reveal the limitations of existing methods when scaled to larger, more complex queries. Interestingly, HProPro with GPT-5 outperforms ODYSSEY on the NBA and Movie domains, which feature tables with thousands of rows (as shown in Table 1), owing to its ability to generate executable programs that directly operate over tables. This result highlights the limitations of ODYSSEY when applied to large-scale tables and aligns with the broader observation that larger table sizes increase the difficulty of table-QA for LLMs Patnaik et al.. The performance gap between GPT-5 and GPT-3.5-turbo (35.6 vs. 20.6 F1 for ODYSSEY and 40.4 vs. 20.3 F1 for HProPro) underscores the importance of advanced LLM reasoning capabilities in handling such challenges. In the retrieval setting, where no gold tables are provided, performance degrades further: the best method (HELIOS + HProPro with GPT-5) attains only 22.6 F1. This sharp decline illustrates the compounded challenge of retrieval and reasoning over heterogeneous corpora.

Table 4: Table-Text QA Accuracy on the SPARTA (Oracle) across multiple domains.

| Method | SPARTA | | | | | | | | | | | | | | | | HybridQA | | | |
| --- | --- | --- | --- | --- | --- | --- | --- | --- | --- | --- | --- | --- | --- | --- | --- | --- | --- | --- | --- | --- |
| | NBA | | | | Movie | | | | Medical | | | | Avg. | | | | | | | |
| | EM | F1 | P | R | EM | F1 | P | R | EM | F1 | P | R | EM | F1 | P | R | EM | F1 | P | R |
| ODYSSEY w/ GPT-3.5-turbo | 9.0 | 15.1 | 26.8 | 14.8 | 20.2 | 23.9 | 33.6 | 24.7 | 6.7 | 22.9 | 33.2 | 21.3 | 12.0 | 20.6 | 31.2 | 20.3 | 32.7 | 42.2 | 42.6 | 44.2 |
| HProPro w/ GPT-3.5-turbo | 11.0 | 13.6 | 16.4 | 13.8 | 22.2 | 27.8 | 29.1 | 29.2 | 15.5 | 19.5 | 20.2 | 19.7 | 16.2 | 20.3 | 21.9 | 20.9 | 21.4 | 25.3 | 25.7 | 26.1 |
| ODYSSEY w/ GPT-5 | 21.2 | 28.4 | **38.4** | 28.1 | 20.4 | 24.2 | 32.9 | 24.3 | **47.5** | **54.2** | **60.3** | **54.2** | **29.7** | 35.6 | **43.9** | 35.5 | 55.3 | 69.5 | 69.3 | **73.5** |
| HProPro w/ GPT-5 | **23.6** | **33.1** | 36.2 | **34.0** | **36.6** | **47.1** | **49.2** | **48.8** | 28.1 | 41.0 | 43.2 | 41.6 | 29.5 | **40.4** | 42.9 | **41.5** | **59.7** | **70.5** | **71.1** | 73.1 |

Table 5: Table-Text QA Accuracy on the SPARTA (Retrieval) across multiple domains.

| Method | SPARTA | | | | | | | | | | | | | | | | OTT-QA | | | |
| --- | --- | --- | --- | --- | --- | --- | --- | --- | --- | --- | --- | --- | --- | --- | --- | --- | --- | --- | --- | --- |
| | NBA | | | | Movie | | | | Medical | | | | Avg. | | | | | | | |
| | EM | F1 | P | R | EM | F1 | P | R | EM | F1 | P | R | EM | F1 | P | R | EM | F1 | P | R |
| HELIOS+FiE Reader | 4.6 | 6.9 | 17.6 | 6.4 | 8.6 | 11.9 | 23.0 | 11.6 | 6.6 | 16.0 | 33.0 | 12.9 | 6.6 | 11.6 | 24.5 | 10.3 | 58.6 | 65.2 | 66.7 | 65.2 |
| HELIOS+HProPro w/ GPT-5 | 14.5 | 19.0 | 24.2 | 18.6 | 17.4 | 21.6 | 28.6 | 21.7 | 13.7 | 27.3 | 31.3 | 27.1 | 15.2 | 22.6 | 28.0 | 22.5 | 47.7 | 56.0 | 57.4 | 56.5 |

We additionally evaluate the FiE Reader Ma et al. (2023), the state-of-the-art fine-tuned reader model on OTT-QA. While FiE Reader surpasses HELIOS + HProPro w/ GPT-5 by 9.2 points on OTT-QA, it lags behind on SPARTA by 11.0 points, showing fine-tuned models fail to generalize to SPARTA's more complex, out-of-domain settings.

## 4.4 ANALYSIS

We conducted a comprehensive analysis of the models' execution results on the SPARTA benchmark. This investigation uncovers several fundamental vulnerabilities in current table-text QA models, pointing to critical directions for future work.

**Models struggle to handle complex multi-hop query structures.** We evaluate Table–Text QA models under various tree-structured query configurations, fixing the number of edges to four: (Depth 1, Breadth 3), (Depth 2, Breadth 2), and (Depth 3, Breadth 1). We also included intermediate shapes with three edges, such as (Depth 1, Breadth 2) and (Depth 2, Breadth 1), to further validate the trend.

As shown in Figure 5a, model performance degrades sharply as either depth or breadth increases. At fixed depth, expanding breadth from (Depth 1, Breadth 1) to (Depth 1, Breadth 3) reduces HProPro and ODYSSEY by 25.2% and 27.5%, respectively. At fixed breadth, increasing depth from (Depth 1, Breadth 1) to (Depth 3, Breadth 1) yields 47.2% and 49.9% declines. Additional comparisons—(Depth 2, Breadth 1) to (Depth 2, Breadth 2), and (Depth 1, Breadth 2) to (Depth 2, Breadth 2)—show consistent degradation, further confirming that both deeper and broader queries cause substantial F1 drops. These findings suggest that existing methods are fundamentally limited in performing tree-structured reasoning over multiple relational paths, regardless of whether complexity arises from depth or breadth.

**Models struggle with analytical operations such as grouping and ordering.** As shown in Figure 5b, both ODYSSEY and HProPro exhibit consistent performance degradation when advanced analytical clauses are present. For queries that include GROUP BY and HAVING clauses, ODYSSEY attains an F1 score of 35.4, whereas HProPro attains 27.1. When ORDER BY and LIMIT are present, the scores are 31.2 for ODYSSEY and 21.4 for HProPro. Aggregation queries show a similar pattern, yielding 28.4 for ODYSSEY and 37.2 for HProPro. Compared with each model's average F1, these analytical scores are markedly lower, indicating weak numerical reasoning, filtering, and ranking capabilities and exposing fundamental limitations in addressing real-world table–text questions. Notably, ODYSSEY performs worst on aggregation queries, whereas HProPro struggles most with ORDER BY and LIMIT.

**Performance drops sharply when unstructured text is required.** As shown in Table 6, the inclusion of table-text cross reasoning leads to a significant decline in performance. HProPro's F1 score drops by 63.9% (from 45.2 to 16.3), while ODYSSEY experiences an even steeper drop of 23.0% (from 39.2 to 28.6). This sharp contrast highlights the difficulty of reasoning over unstructured passages in conjunction with structured tables. Although both models perform moderately well when queries rely only on tabular data, they consistently fail to retrieve and integrate relevant textual spans when external context is present. These failures indicate that current Table–Text QA models lack robust cross-modal alignment and semantic grounding, limiting their effectiveness in real-world scenarios that demand joint reasoning over heterogeneous data sources. To further support our findings, we

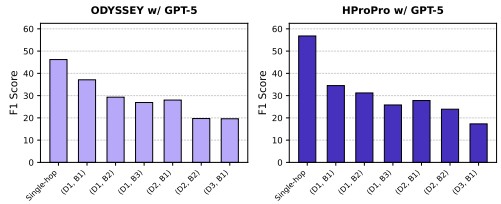
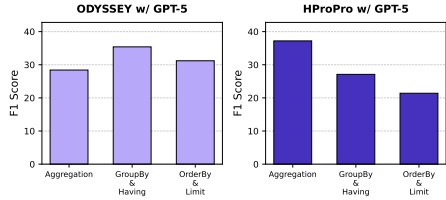

(a) F1 scores across tree configs (D: Depth, B: Breadth)  (b) F1 scores across analytical operations

Figure 5: Comparison of F1 scores across different configurations.

Table 6: Performance Comparison With and Without Text Data in Table-Text QA.

| Method | Setting | EM | F1 | P | R |
|---|---|---|---|---|---|
| ODYSSEY w/ GPT-5 | Table-Text Cross Reasoning | 23.9 | 28.6 | 36.3 | 28.4 |
| | Table-only Reasoning | 32.0 | 39.2 | 49.5 | 38.8 |
| HProPro w/ GPT-5 | Table-Text Cross Reasoning | 11.9 | 16.3 | 17.2 | 16.7 |
| | Table-only Reasoning | 29.2 | 45.2 | 46.9 | 46.4 |

include supplementary analyses in the appendix: an ablation study on nesting types (Appendix G) and an error case analysis (Appendix J).

## 5 CONCLUSION

In summary, we present SPARTA, a benchmark generation framework that rectifies the three critical shortcomings of existing Table–Text QA resources—shallow question design, annotation noise, and toy-scale tables—by (i) unifying heterogeneous evidence inside a reference fact database, (ii) generating logically deep, human-like SQL nested queries whose hop count and analytical operations are explicitly controlled through a provenance-guided LLM pipeline, and (iii) verbalising them into natural-language questions using an LLM-based SQL-to-text model, with lightweight human validation for fluency and correctness. On SPARTA, state-of-the-art models that reach over 70 F1 on HybridQA or over 50 F1 on OTT-QA drop by more than 30 F1 points, exposing fundamental weaknesses in current cross-modal reasoning.

## 6 LIMITATIONS AND FUTURE WORK

SPARTA currently focuses on the Table–Text setting. Future work will extend it to multimodal inputs like images and videos by using vision–language models to summarize visuals into atomic statements, normalizing them into grounding tables, and merging with the existing fact database. Since these tuples follow the same schema as $\mathcal{D}$, the query-generation pipeline (§3.3) applies unchanged. A complete multimodal extension, including dataset collection, schema design, and evaluation, is planned for future research.

## 7 ACKNOWLEDGMENTS

This work was partly supported by the National Research Foundation of Korea(NRF) grant funded by the Korea government(MSIT) (RS-2025-00517736, 30%), Institute of Information & communications Technology Planning & Evaluation (IITP) grant funded by the Korea government(MSIT) (No. RS-2024-00509258, Global AI Frontier Lab, 50%)(No. RS-2024-00454666, Developing a Vector DB for Long-Term Memory Storage of Hyperscale AI Models, 10%), and Basic Science Research Program through the National Research Foundation of Korea Ministry of Education (No. RS-2024-00415602, 10%)

**Reproducibility Statement** We include prompt examples for provenance-based refinement, realistic-structure enforcement, and automatic naturalness evaluation in Appendix N. Details of the experimental setup are provided in Section 4.1.

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

# A  CROSS-DATASET ANNOTATION AUDIT

| Dataset | Any error | Error breakdown (within erroneous samples) | | |
|---|---|---|---|---|
| | % of all samples | Redundant modality | Incomplete answer set | Incorrect / unanswerable |
| HYBRIDQA | 21% | 52.4% | 23.8% | 23.8% |
| MULTIHIERITT | 26% | 15.4% | 30.8% | 53.8% |
| TAT-QA | 30% | 96.7% | 0.0% | 3.3% |
| FINQA | 17% | 41.2% | 0.0% | 58.8% |
| SPARTA | 0% | 0.0% | 0.0% | 0.0% |

Table 7: Audit of 100 randomly sampled dev examples from each dataset. "Any error" shows the fraction of all samples containing at least one error. The breakdown columns report the relative distribution among erroneous samples.

# B  SUPPORTED NESTED QUERY PATTERNS

SPARTA synthesizes queries for each of the four primary nesting patterns Kim (1982) commonly observed in real-world SQL, as illustrated in Fig 6.

Table 8: Nested–query patterns.

| Type | Inner aggreg. | Correlation | Typical intent / example |
|---|---|---|---|
| Type–N | ✗ | ✗ | *Pure set membership.* Outer block tests whether a value belongs to the set returned by a non-correlated subquery (e.g., `WHERE x IN (SELECT ...)`). |
| Type–A | ✓ | ✗ | *Aggregate comparison.* Inner block computes an aggregate such as `AVG` or `MAX` and the result is compared with each outer tuple (e.g., `salary > (SELECT AVG(salary) FROM ...)`). |
| Type–J | ✗ | ✓ | *Correlated filtering.* Inner query references attributes of the outer block without aggregation (e.g., `EXISTS (SELECT 1 FROM Items i WHERE i.order_id = o.id)`). |
| Type–JA | ✓ | ✓ | *Correlated aggregate comparison.* Inner query both correlates with the outer block and aggregates its own rows before the comparison (e.g., `EXISTS (SELECT 1 FROM Items i WHERE i.order_id = o.id GROUP BY ...  HAVING SUM(i.qty) > o.limit)`). |

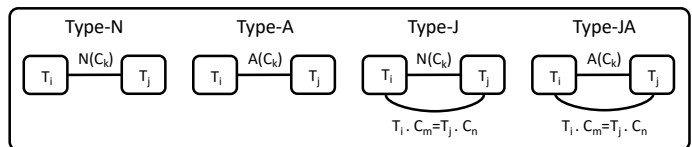

Figure 6: Four primary nesting patterns—type (N, A, J, JA) queries of depth 1. Each consists of an outer block ($T_i$) and an inner block ($T_j$). Arcs labeled 'A' indicate aggregation in the inner `SELECT`; straight arcs 'N' denote set-inclusion predicates; curved arcs denote join predicates.

# C   SCHEMA OF REFERENCE-FACT DATABASE

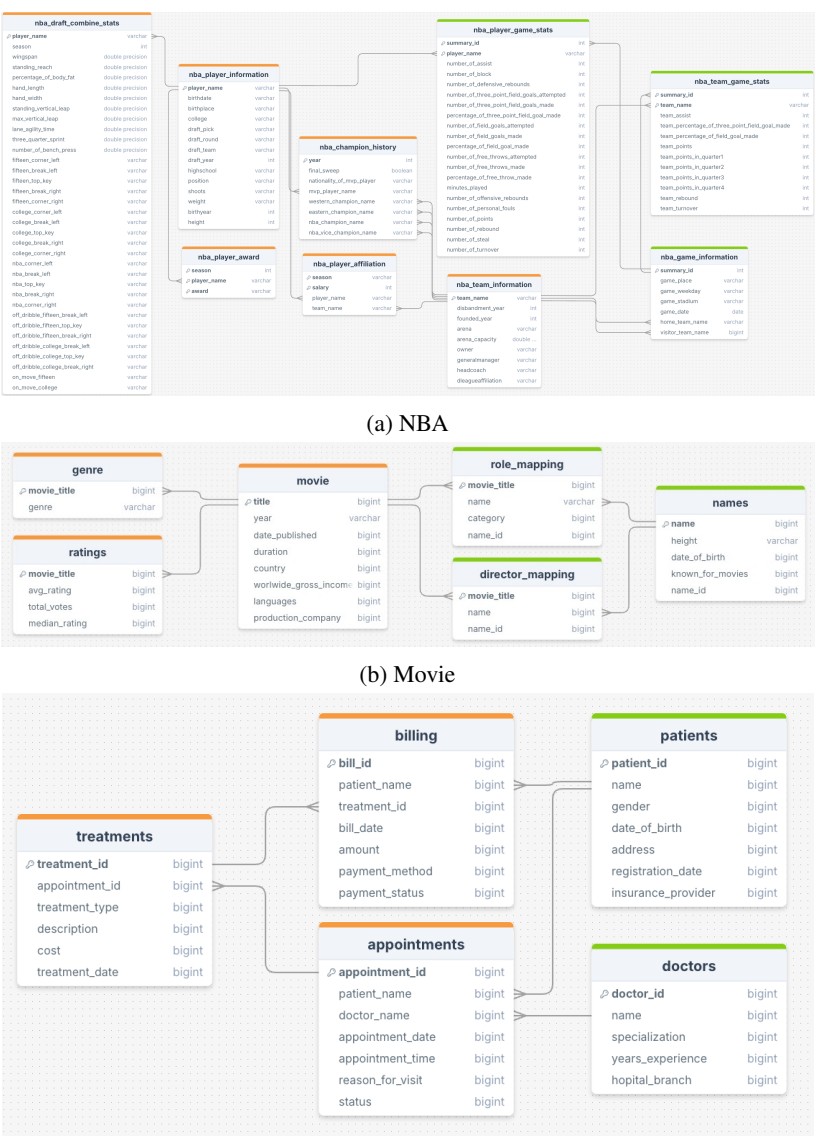

(a) NBA

(b) Movie

(c) Medical

Figure 7: Schemas of the reference-fact databases used in SPARTA across three domains. Each database consists of two complementary types of tables: source tables $S_T$ (orange) from public datasets (e.g., NBA player salaries, movie metadata, medical records) and grounding tables $G_T$ (green) encoding atomic facts extracted from textual passages.

# D  TABLE-LEVEL STATISTICS OF THE REFERENCE-FACT DATABASE

Table 9: Row/Column Statistics of All Tables in the Sparta Benchmark

| Domain | Table Name | # Columns | # Rows |
|--------|-----------|-----------|--------|
| nba | nba_draft_combine_stats | 35 | 772 |
| | nba_player_information | 14 | 4596 |
| | nba_player_award | 3 | 236 |
| | nba_champion_history | 8 | 69 |
| | nba_player_affiliation | 4 | 13980 |
| | nba_team_information | 9 | 30 |
| | nba_player_game_stats | 21 | 45640 |
| | nba_team_game_stats | 12 | 12750 |
| | nanba_game_informtaion | 7 | 6665 |
| imdb | genre | 2 | 14418 |
| | ratings | 4 | 7872 |
| | movie | 8 | 7872 |
| | role_mapping | 4 | 15336 |
| | director_mapping | 3 | 3800 |
| | names | 5 | 25617 |
| medical | treatments | 6 | 200 |
| | billing | 7 | 200 |
| | appointments | 7 | 200 |
| | patients | 7 | 50 |
| | doctors | 5 | 10 |

Table 9 provides an overview of the structural statistics for all tables in the SPARTA benchmark, including the number of columns and rows per table, grouped by domain (NBA, IMDB, and Medical). These metrics highlight the scale and diversity of the reference-fact database used for evaluation.

# E  BENCHMARK CONFIGURATION

Table 10: Benchmark Configuration: SQL Operator Coverage, Query-Shape/Size Distribution.

| Query Shape and Size Distribution (%) | | | | | | | |
|---|---|---|---|---|---|---|---|
| Non-nested | (Depth 1, Breadth 1) | (Depth 1, Breadth 2) | (Depth 1, Breadth 3) | (Depth 2, Breadth 1) | (Depth 2, Breadth 2) | (Depth 3, Breadth 1) | Total |
| 45.5 | 9.1 | 9.1 | 9.1 | 9.1 | 9.1 | 9.1 | 100.0 |

| SQL Operator Presence (%) | | | | | | |
|---|---|---|---|---|---|---|
| WHERE | GROUP BY | HAVING | ORDER BY | LIMIT | AGGREGATION | |
| 100.0 | 15.3 | 3.4 | 7.7 | 4.5 | 50.0 | |

| Nested Predicate Type Presence in Nested Query (%) | | | |
|---|---|---|---|
| Type-N | Type-A | Type-J | Type-JA |
| 57.8 | 64.3 | 32.4 | 15.2 |

# F  GENERATION COST ANALYSIS

## F.1  COST ANALYSIS ACROSS LLM SCALES

To demonstrate that our provenance-based refinement is effective regardless of the LLM's size, we conducted additional experiments comparing generation costs across LLMs of varying sizes. Specifically, in addition to the Llama-3.1-70B-Instruct model evaluated in the manuscript, we measured generation costs using a smaller-parameter LLM (gpt-oss-20B) and a larger-parameter LLM (gpt-oss-120B).

Table 11 shows that Post-Order + Prov is the most cost-effective approach across all LLM variants, completing with 4854, 4,722, and 3831 calls for the respective models, while cutting call volume by 18.8%, 42.8%, and 54.5% versus vanilla Post-Order, and by 64.7%, 66.2%, and 65.7% versus One-Shot-k. These results indicate that disciplined post-order construction combined with provenance-driven repair minimizes redundant generations independent of the LLM's scale.

Note that the "Ideal Calls" metric represents the number of LLM calls required if every step succeeds. It varies slightly due to probabilistic clause inclusion in SQL query generation (based on predefined per-clause probabilities). As shown in the table, the variance is very small when we repeat experiments three times per method.

| LLM | Method | Ideal Calls | Total Calls |
|---|---|---|---|
| | One-Shot–k | 2661 | 13736 |
| gpt-oss-20B | Post-Order (no provenance) | 3073 | 5977 |
| | Post-Order + Prov | 3146 | 4854 |
| | One-Shot–k | 2664 | 13962 |
| Llama-3.1-70B-Instruct | Post-Order (no provenance) | 3104 | 8253 |
| | Post-Order + Prov | 3074 | 4722 |
| | One-Shot–k | 2621 | 11225 |
| gpt-oss-120B | Post-Order (no provenance) | 3152 | 8425 |
| | Post-Order + Prov | 3145 | 3831 |

Table 11: Generation cost comparison across LLM sizes for three refinement strategies.

## F.2 COST ANALYSIS ON QUERY SHAPE AND SIZE

Figure 8 contrasts LLM usage for *star* and *chain* query trees as their size grows from one to three nested predicates. Here, *star* queries fix depth to 1 while increasing breadth (number of branches), whereas *chain* queries fix breadth to 1 while increasing depth (number of nested levels). The size thus reflects how many nested predicates are added along either the breadth or depth dimension. For star shapes, one-shot generation quickly becomes prohibitive, ballooning to $17\times$ the ideal call count when the hub size reaches three. Building the same queries in post-order slashes that overhead to $3.2\times$; provenance repair trims it further to $1.6\times$. Chains tell a different story: because their natural construction order already matches post-order, one-shot and post-order costs are similar, yet provenance still removes 30–40 % of redundant calls at every depth. Branching structures profit most from post-order generation, while provenance-guided repair is a universally cheap "insurance policy" that cuts waste regardless of query shape.

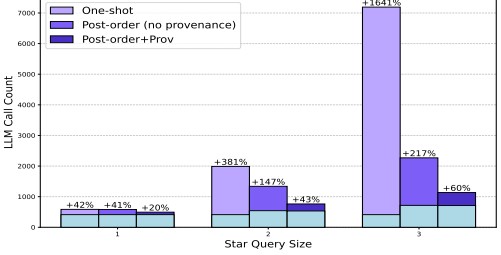
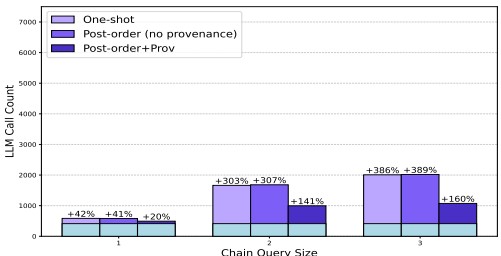

(a) LLM Call Count as Star-Query Size Increases.  (b) LLM Call Count as Chain-Query Size Increases.

Figure 8: Generation cost for varying (a) star-query size and (b) chain-query size. Sky-blue bars mark ideal LLM calls, and the labels above each bar represent the actual excess percentage.

Beyond structural complexity, we also measured the average number of LLM calls required to generate a single query as the number of accessed tables increases. Table 12 shows increases of +2.6 calls (from 1 to 2 tables), +3.9 (from 2 to 3), and +2.2 (from 3 to 4), indicating near-linear growth.

Table 12: Average LLM calls per query as the number of accessed tables increases.

| # of Accessed Tables | Average LLM Calls per Query |
|---|---|
| 1 | 2.3 |
| 2 | 4.9 (+ 2.6) |
| 3 | 8.8 (+ 3.9) |
| 4 | 11.0 (+ 2.2) |

## G   ABLATION STUDY ON NESTING TYPES

For a deeper analysis, we examined the best-performing system on the SPARTA Benchmark—HProPro with GPT-5—by breaking down its performance across query-nesting types: plain nesting (N), nesting with aggregates (A), nesting with correlated joins (J), and nesting that combines correlated joins + aggregates (JA). Results are shown below.

Table 13: F1 scores of HProPro (w/ GPT-5) across different nesting types. Percentage values indicate relative change from the overall average (34.5).

| Nesting Type | F1 (HProPro w/ GPT-5) |
|---|---|
| Type-N | 40.0  (+15.9%) |
| Type-A | 33.7  (−2.3%) |
| Type-J | 30.8  (−10.7%) |
| Type-JA | 25.6  (−25.8%) |
| Total | 34.5 |

As shown in Table 13, F1 falls steadily as structural (correlated joins) and analytical (aggregates) complexity increases, with the largest drop when both factors are present (Type JA). This ablation study underscores that correlated joins and aggregates are the model's primary pain points.

## H   ANALYSIS ON NEGATION AND RANGE REASONING

To better understand which logical operators and conditions most challenge SPARTA models, we conduct an ablation study focusing on two key query categories: *negation* and *numeric range* conditions. These categories capture a large portion of structural breadth in SPARTA and represent frequent sources of model errors.

We evaluate HProPro with GPT-5 on queries containing explicit negation operators (`NOT LIKE`, `NOT EXISTS`, `NOT IN`, `<>`) as well as numeric range operators (`>`, `<`, `>=`, `<=`). Table 14 presents the results.

Table 14: F1 scores of HProPro (w/ GPT-5) on negation and range queries.

| Query Category | F1 (HProPro w/ GPT-5) |
|---|---|
| Negation (`NOT LIKE`, `NOT EXISTS`, `NOT IN`, `<>`) | 28.7 (−28.3%) |
| Range (`>`, `<`, `>=`, `<=`) | 32.9 (−18.6%) |
| SPARTA (Oracle) | 40.4 |

Both categories show notable degradation compared to the overall SPARTA score, with negation queries dropping by 28.3% and range queries by 18.6%. These findings indicate that logical negation and numeric range reasoning remain significant bottlenecks.

In addition to this quantitative breakdown, we selected representative samples from both query types and conducted a qualitative error analysis, shown in Figure 9. As illustrated in Figure 9(a), negation reasoning presents a recurring challenge. The example query requires identifying players for whom *there is no record indicating* that they (i) are a Center with height $> 90$, (ii) were born after 1970, and (iii) were drafted "9th overall." All three criteria fall under the scope of a single negated condition. However, the model applies negation only to the first clause ("Center with height $> 90$") while incorrectly treating the remaining clauses ("born after 1970," "drafted 9th overall") as independent positive filters. This partial scoping of negation leads the model to misinterpret the logical structure and include players who should have been excluded.

A similar pattern is observed for range reasoning, as shown in Figure 9(b). The gold query requires identifying teams whose arena capacity exceeds the maximum capacity among teams that (i) were founded before 1970 and (ii) have capacities between 20,000 and 21,711. Although the model correctly computes the maximum capacity among pre-1970 teams, it fails to apply the upper bound constraint (`arena_capacity < 21,711`) during the final filtering stage. As a result, the

predicted code returns teams that satisfy the lower-bound conditions and the dynamic threshold but violate the required upper-bound range condition, producing an incorrect answer.

**Question:** What is the maximum number of points scored in a single game by player who, in that game, recorded more than 5 rebounds and more than 5 steals, and weight is less than 180lb and for whom there is no record indicating that they are a Center with height greater than 90, born after 1970, and drafted '9th overall'?
**Answer**: 19

**Gold Code:**
```
def solve(table) -> str:
    for row in table:
        if not (row[weight][0] < 180):
            continue
        if ('Center' in row[position][0] and row[height][0] > 90 and
            row[birthyear][0] > 1970 and row[draft_pick][0] == '9th overall'):
            continue
        passage_links = row[player_name][1]
        # ... extract the points with >5 rebounds and >5 steals ...
    return max_points
```

**Predicted Code:**
```
def solve(table) -> str:
    max_points = []
    for row in table:
        if not (row[weight][0] < 180):
            continue
        if ('Center' in row[position][0] and row[height][0] > 90):
            continue
        if not (row[birthyear][0] > 1970 and row[draft_pick][0] == '9th overall'):
            continue
        passage_links = row[player_name][1]
        # ... extract the points with >5 rebounds and >5 steals ...
    return max_points
```
(a)

**Question:** What is the highest team points in a game for any team whose arena capacity is above the maximum arena capacity of any team founded before 1970, with an arena capacity above 20000, and owned by someone other than Paul Allen, and with an arena capacity below 21711, and which scored more than 100 points and rebounded more than 40 times in a game?
**Answer**: 127

**Gold Code:**
```
def solve(table) -> str:
    max_before_1970 = None
    for row in table:
        if (row['founded_year'][0] < 1970) and (row['owner'][0] != 'Paul Allen') and
            (row['arena_capacity'][0] > 20000 and row['arena_capacity'][0] < 21711):
            if row['arena_capacity'][0] > max_before_1970:
                max_before_1970 = row['arena_capacity'][0]
    for row in table:
        if arena_capacity > max_before_1970:
            passage_link = row['team_name'][1]
            # ... extract the points with >100 points and >40 rebounds ...
    return max_points max_before_1970
```

**Predicted Code:**
```
def solve(table) -> str:
    max_before_1970 = None
    for row in table:
        if (row['founded_year'][0] < 1970) and (row['owner'][0] != 'Paul Allen') and
            (row['arena_capacity'][0] > 20000 and row['arena_capacity'][0] < 21711):   missing!
            if row['arena_capacity'][0] > max_before_1970:
                max_before_1970 = row['arena_capacity'][0]
    for row in table:
        if arena_capacity > max_before_1970:
            passage_link = row['team_name'][1]
            # ... extract the points with >100 points and >40 rebounds ...
    return max_points
```
(b)

Figure 9: Illustration of representative error cases where the model fails to correctly answer. (a) Negation reasoning error. (b) Range reasoning error.

## I  ABLATION STUDY ON ROBUSTNESS TO LINGUISTIC VARIABILITY

We further study the robustness of table-text qa models to linguistic variability by evaluating performance under human-verified rephrased questions. Specifically, we sampled 100 queries from our benchmark and had them manually rephrased by human annotators, ensuring the core semantic meaning and the correct answer were preserved. We then evaluated the HProPro model with GPT-5 on both the original and the rephrased sets of queries.

Table 15: F1 scores of HProPro (w/ GPT-5) on original and human-verified rephrased queries.

| Query Set | F1 (HProPro w/ GPT-5) |
|---|---|
| Original Questions | 45.22 |
| Rephrased Questions | 45.02  $(-0.44\%)$ |

As shown in Table 15, the F1 score dropped from 45.22 on the original queries to 45.02 on the rephrased versions, amounting to a negligible decrease of 0.44%. This finding indicates that the model is highly robust to surface-level linguistic variations. We have incorporated the details of this new experiment and its results into the Appendix I of our revised manuscript to strengthen our robustness analysis.

## J  ERROR CASE ANALYSIS

We conduct an analysis of the errors encountered by Table-Text QA models on randomly sampled sets of 100 examples each for SPARTA (Oracle) and SPARTA (Retrieval), as illustrated in Fig 10. Representative error types, along with their frequencies and causal interpretations, are summarized below.

**Relevant data missing.**   This was the most frequent category of failure, where the model failed to identify all the necessary information to correctly answer the question. SPARTA poses increased

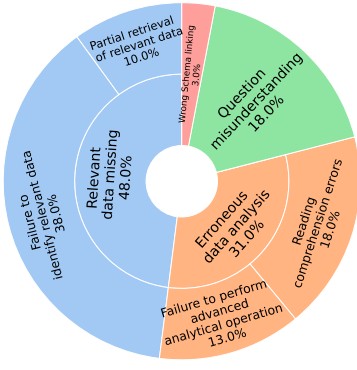 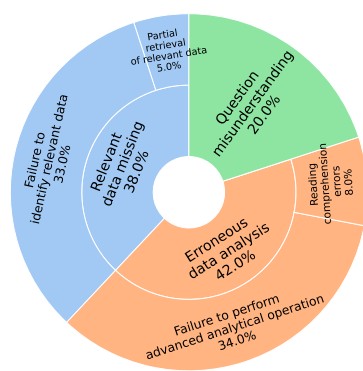

(a) Oracle Setting          (b) Retrieval Setting

Figure 10: Statistics of errors. For detailed descriptions and examples of each error category, see Appendix J.

demands for multi-hop reasoning across table and text sources, which existing methods often struggle with:

- **Partial retrieval of relevant data:** The model identifies only a subset of the necessary sources, resulting in incomplete answers. As illustrated in Figure 11, the model was expected to return both 62 and 53 as the field goal percentages for the `Dallas Mavericks` and `New York Knicks`, respectively, but failed to do so.

- **Failure to identify relevant data:** The model does not identify crucial supporting data, leading to either no answer or an incorrect one. For example, in questions requiring information from both `nba_player_information` and `nba_player_award`, the model may access only the former, overlooking the award records, and consequently returning an incorrect answer.

**Erroneous data analysis.** Compared to prior benchmarks, `SPARTA` introduces more complex analytical requirements that reveal limitations in model capabilities:

- **Failure to perform advanced analytical operations:** The model struggles with applying operations such as aggregations (e.g., `COUNT`, `MAX`) or executing multi-table joins correctly. These operations require precise alignment of relational structures and logic, which is frequently mishandled.

- **Reading comprehension errors:** The model incorrectly interprets textual information, leading to erroneous answers. For instance, in a case where the question asks for the Nuggets' field goal percentage, the model erroneously extracts "37%" from the sentence "`Nuggets held Sacramento to just 37 percent from the field,`" misattributing Sacramento's statistic to the Nuggets. See Figure 11 for a detailed example of this error.

**Question misunderstanding.** These errors arise from incorrect interpretation of the question intent or constraints. Representative cases include failing to restrict answers to players who played only as `point_guard`, and instead including players who played `point_guard` along with other positions, misidentifying the relevant time frame (e.g., using 2017 instead of 2016–17), introducing constraints not specified in the question, or omitting key conditions necessary to derive the correct answer.

**Schema linking errors.** This category involves incorrect associations between the question and the schema elements, such as tables or columns. For instance, when asked to retrieve the name of the head coach, the model fails to identify the `headcoach` column in the `nba_team_information` table as relevant, thereby omitting necessary information from the final prediction.

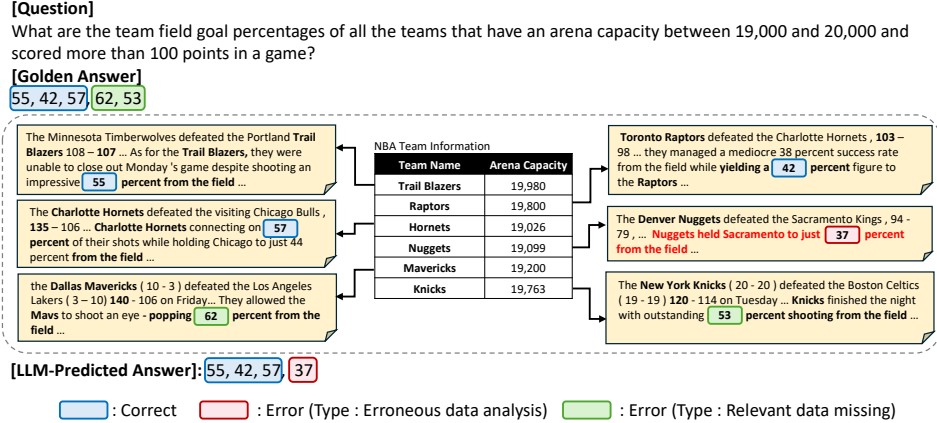

Figure 11: Illustration of a representative error case where the model fails to correctly answer.

## K    SOFTWARE AND DATA LICENSES

The licenses for the software and datasets used in this paper are as follows:

- LLaMA 3.1-70B-Instruct: LLaMA 3.1
- OTT-QA: MIT License
- HybridQA: MIT License

All software and datasets were used strictly for research purposes and were not utilized in any non-research contexts, particularly for commercial applications.

## L    AI ASSISTANTS

We used ChatGPT-4o OpenAI to debug code efficiently, quickly identifying and resolving errors in our implementations. Additionally, we used it for rephrasing sentences in our writing to improve clarity and readability.

## M    REPRESENTATIVE EXAMPLES FROM OUR SPARTA BENCHMARK

Table 16: 20 representative examples from SPARTA, each consisting of a domain, a natural language question, its corresponding SQL query, and the answer.

| Row Type | Content |
| --- | --- |
| **Domain** | NBA |
| **Question** | Which player won the NBA MVP award in the 1986 season? |
| **SQL** | `SELECT player_name FROM nba_player_award WHERE season = 1986 AND award = 'nba mvp'` |
| **Answer** | Larry Bird |
| **Domain** | NBA |
| **Question** | What are the names of the players who scored more than 15 points and rebounded more than 5 times in a game? |
| **SQL** | `SELECT player_name FROM nba_player_game_stats WHERE number_of_points > 15 AND number_of_rebound > 5` |
| **Answer** | Langston Galloway, Quincy Acy, Larry Nance Jr., ... |
| **Domain** | Movie |
| **Question** | In which movies did Riteish Deshmukh act? |
| **SQL** | `SELECT movie_title FROM role_mapping WHERE category = 'actor' AND name = 'Riteish Deshmukh'` |
| **Answer** | Marjaavaan, Mauli |
| **Domain** | Movie |
| **Question** | What is the total number of movies with a median rating greater than 5 and an average rating greater than 5.5? |
| **SQL** | `SELECT COUNT(movie_title) AS total_movies FROM ratings WHERE median_rating > 5 AND avg_rating > 5.5` |
| **Answer** | 4877 |
| **Domain** | NBA |
| **Question** | Which Western Conference teams faced the Celtics more than once in the Finals? |
| **SQL** | `SELECT  western_champion_name  FROM nba_champion_history WHERE nba_champion_name = 'Celtics' GROUP BY western_champion_name HAVING COUNT(western_champion_name ) > 1` |
| **Answer** | Rockets, Lakers |
| **Domain** | Medical |
| **Question** | What is the maximum years of experience of a pediatrician at Central Hospital? |
| **SQL** | `SELECT MAX(years_experience) FROM doctors WHERE hospital_branch = 'Central Hospital' AND specialization = 'Pediatrics'` |
| **Answer** | 28 |
| **Domain** | NBA |
| **Question** | What is the highest salary of Kevin McHale while playing for the Celtics? |
| **SQL** | `SELECT MAX(salary) FROM nba_player_affiliation WHERE player_name = 'Kevin McHale' AND team_name = 'Celtics'` |
| **Answer** | 3,500,000 |
| **Domain** | NBA |
| **Question** | Which Point Guards, drafted between 2000 and 2005, had more than 4 three-pointers, more than 8 field goals and more than 1 steal in a game? |

*(continued on next page)*

| Row Type | Content (continued) |
|----------|---------------------|
| **SQL** | `SELECT player_name FROM nba_player_game_stats WHERE player_name IN (SELECT player_name FROM nba_player_information WHERE position = 'Point Guard' AND draft_year BETWEEN 2000 AND 2005) AND number_of_three_point_field_goals_made > 4 AND number_of_field_goals_made > 8 AND number_of_steal > 1` |
| **Answer** | Chris Paul |
| **Domain** | NBA |
| **Question** | Which NBA players who were drafted in the first round and play the center position have a salary of over 1 million in the 2016-17 season? |
| **SQL** | `SELECT player_name FROM nba_player_information WHERE player_name IN (SELECT player_name FROM nba_player_affiliation WHERE salary > 1000000 AND season = '2016-17') AND draft_round = '1st round' AND position = 'Center'` |
| **Answer** | Alex Len, Al Horford, Andre Drummond, ... |
| **Domain** | Medical |
| **Question** | What are the names of patients who have an appointment with a doctor who works at the central hospital and has more than 20 years of experience? |
| **SQL** | `SELECT patient_name FROM appointments WHERE doctor_name IN (SELECT name FROM doctors WHERE hospital_branch = 'Central Hospital' AND years_experience > 20)` |
| **Answer** | Alex Smith, Alex Aiden Moore, Emily Miller, ... |
| **Domain** | NBA |
| **Question** | What are the years of birth of the players who have a lane agility time of more than 11.5 seconds, a three quarter sprint of less than 3.35 seconds, more than 10 field goals made and more than 8 rebounds in a game? |
| **SQL** | `SELECT birthyear FROM nba_player_information WHERE player_name IN (SELECT player_name FROM nba_draft_combine_stats WHERE lane_agility_time > 11.5 AND three_quarter_sprint < 3.35) AND player_name IN (SELECT player_name FROM nba_player_game_stats WHERE number_of_field_goals_made > 10 AND number_of_rebound > 8) GROUP BY birthyear` |
| **Answer** | 1984, 1985, 1989, ... |
| **Domain** | Movie |
| **Question** | Which movies, starring Vincent D Onofrio as an actor, have an average rating greater than 5 and a median rating of 6, excluding 'Kolonya Cumhuriyeti'? |
| **SQL** | `SELECT title FROM movie WHERE title IN (SELECT movie_title FROM role_mapping WHERE category = 'actor' AND name = 'Vincent D Onofrio' AND movie_title = title) AND title IN (SELECT movie_title FROM ratings WHERE avg_rating > 5 AND median_rating = 6 AND movie_title <> 'Kolonya Cumhuriyeti')` |
| **Answer** | CHIPS, In Dubious Battle |
| **Domain** | NBA |
| **Question** | Who are the top 5 centers drafted in the 1st round, who have won the dpoy award after 2000, and who have earned more than 2 million dollars in the 2004-05 season, sorted by their draft year in descending order and birth year in ascending order? |

| Row Type | Content (continued) |
|---|---|
| **SQL** | `SELECT player_name FROM nba_player_information WHERE player_name IN (SELECT player_name FROM nba_player_affiliation WHERE salary > 2000000 AND season = '2004-05') AND player_name IN (SELECT player_name FROM nba_player_award WHERE season > 2000 AND award = 'dpoy') AND position = 'Center' AND draft_round = '1st round' ORDER BY draft_year DESC, birthyear ASC LIMIT 5` |
| **Answer** | Dwight Howard, Ben Wallace, ... |
| **Domain** | Medical |
| **Question** | Find the addresses of male patients born after January 1, 1980, who have MedCare Plus insurance and have made payments that exceed the average failed payments greater than 2500. |
| **SQL** | `SELECT address FROM patients WHERE name IN (SELECT patient_name FROM billing WHERE amount > (SELECT AVG(amount) FROM billing WHERE payment_status = 'Failed' AND amount > 2500)) AND date_of_birth > '1980-01-01' AND gender = 'M' AND insurance_provider = 'MedCare Plus'` |
| **Answer** | 123 Elm St, 789 Pine Rd, ... |
| **Domain** | NBA |
| **Question** | Which NBA players, who are centers and taller than the average height of point guards drafted after 1990, have more than 8 rebounds in a game? |
| **SQL** | `SELECT player_name FROM nba_player_game_stats WHERE player_name IN (SELECT player_name FROM nba_player_information WHERE height > (SELECT AVG(height) FROM nba_player_information WHERE position = 'Point Guard' AND draft_year > 1990) AND position = 'Center') AND number_of_rebound > 8` |
| **Answer** | Alex Len, Al Horford, Andre Drummond, ... |
| **Domain** | Medical |
| **Question** | What are the names of female patients who registered after 2021-09-02 and have billed amounts greater than the average amount of failed payments over 2500? |
| **SQL** | `SELECT name FROM patients WHERE name IN (SELECT patient_name FROM billing WHERE amount > (SELECT AVG(amount) FROM billing WHERE payment_status = 'Failed' AND amount > 2500)) AND gender = 'F' AND registration_date > '2021-09-02'` |
| **Answer** | Emily Jones, Laura Aiden Davis, ... |
| **Domain** | Movie |
| **Question** | How many movies starring John Abraham have a median rating above 5 and average rating above 4? |
| **SQL** | `SELECT COUNT(title) AS number_of_movies FROM movie WHERE title IN (SELECT movie_title FROM role_mapping WHERE category = 'actor' AND name = 'John Abraham' GROUP BY movie_title) AND title IN (SELECT movie_title FROM ratings WHERE median_rating > 5 AND avg_rating > 4)` |
| **Answer** | 1 |
| **Domain** | NBA |
| **Question** | What is the maximum height of the Lakers players who play as center, were drafted after 1995 and have a salary greater than the highest salary of the Suns and greater than 20,000,000? |
| **SQL** | `SELECT MAX(height) FROM nba_player_information WHERE player_name IN (SELECT player_name FROM nba_player_affiliation WHERE salary > (SELECT MAX(salary) FROM nba_player_affiliation WHERE team_name = 'Suns') AND salary > 20000000 AND team_name = 'Lakers') AND position = 'Center' AND draft_year > 1995` |

| Row Type | Content (continued) |
| --- | --- |
| **Answer** | 84 |
| **Domain** | NBA |
| **Question** | What are the names of the teams that scored more than the highest points scored by the Thunder when they scored more than 25 points in the first quarter and scored more than the highest points scored by teams that scored more than 100 points and had a three point field goal percentage of more than 30 and have an arena capacity of more than 20,000 and are not the Pistons? |
| **SQL** | ```
SELECT team_name FROM nba_team_game_stats
WHERE team_points > (SELECT MAX(team_points)
FROM nba_team_game_stats WHERE team_name =
'Thunder' AND team_points_in_quarter1 > 25) AND
team_points > (SELECT MAX(team_points) FROM
nba_team_game_stats WHERE team_points > 100 AND
team_percentage_of_three_point_field_goal_made > 30) AND
team_name IN (SELECT team_name FROM nba_team_information
WHERE arena_capacity > 20000 AND team_name <> 'Pistons')
``` |
| **Answer** | Bulls |
| **Domain** | Movie |
| **Question** | Which movies directed by Vivek Athreya have a median rating greater than 5 with more than 100 total votes, and do not feature Matt Smith as an actor? |
| **SQL** | ```
SELECT title FROM movie WHERE title IN (SELECT
T2.movie_title FROM director_mapping AS T2 WHERE T2.name
= 'Vivek Athreya' AND movie.title = T2.movie_title) AND
NOT title IN (SELECT movie_title FROM role_mapping WHERE
category = 'actor' AND name = 'Matt Smith') AND title IN
(SELECT movie_title FROM ratings WHERE median_rating > 5
AND total_votes > 100 GROUP BY movie_title)
``` |
| **Answer** | Brochevarevarura, Mental Madhilo |

## N    PROMPT TEMPLATES

We define a suite of prompt templates that guide LLMs to generate executable, semantically coherent SQL queries. Prompts are organized into three categories, with an NBA domain example provided; for other domains, only domain-specific tokens are swapped (e.g., replacing "NBA" with "Movie").

**Clause-Level Generation.** Templates for generating individual SQL clauses in canonical order:

- **SELECT** (non-aggregate, aggregate)
- **FROM**
- **WHERE**
- **GROUP BY**
- **HAVING**
- **ORDER BY**
- **LIMIT**

**Nested Predicate Construction.** Templates for building multi-hop queries via nested predicates:

- **Inner Query Selection**
- **FROM Clause for Outer Block**
- **Nested Predicate Generation**: Type-N, Type-A, Type-J, Type-JA

**Refinement and Evaluation.** Templates to improve query validity and assess realism:

- **Provenance-Based Refinement** for repairing empty-result queries
- **Naturalness Evaluation** to assess relevance and intent clarity

---

**WHERE Clause Generation**

You are both an NBA fan and an SQL expert. Given the provided database and the generated clauses, generate a WHERE clause that reflects authentic NBA-related curiosity.

Ensure the following requirements:

- Output Structure: Return a JSON object containing a single key, `"where"`, with its value being a WHERE clause.
- Ensure NBA Fan Relevance: Generate the WHERE clause that aligns naturally with realistic and meaningful queries that NBA fans are likely to ask.
- Maintain Specificity and Clarity of Intent: Generate the WHERE clause that is well-defined, avoiding overly vague or artificially complex queries.
- Align with Generated Clauses: Ensure that the WHERE clause maintains logical consistency with previously generated clauses, preserving semantic coherence.
- Ensure Synthetic Correctness: Generate the WHERE clause that is syntactically correct and executable on the provided database.

**IMPORTANT**: Do not generate conditions for `NULL` or `None` values. Also, avoid generating filter conditions that duplicate any existing filters.

Database: `{database}`

Generated Clauses: `{generated_clauses}`

Return the results in a FLAT JSON format.
**DO NOT** include any explanations or notes in the output. **ONLY** return JSON.

---

**GROUP BY Clause Generation**

You are both an NBA fan and an SQL expert. Given the provided database and the generated clauses, generate a GROUP BY clause that reflects authentic NBA-related curiosity.

Ensure the following requirements:

- Output Structure: Return a JSON object containing a single key, `"group"`, with its value being a GROUP BY clause. The GROUP BY clause should include a single column.
- Ensure NBA Fan Relevance: Generate the GROUP BY clause that aligns naturally with realistic and meaningful queries that NBA fans are likely to ask.
- Align with Generated Clauses: Ensure that the GROUP BY clause maintains logical consistency with previously generated clauses, preserving semantic coherence.
- Ensure Synthetic Correctness: Generate the GROUP BY clause that is syntactically correct and executable on the provided database.

**IMPORTANT**: Do not group by any column whose value is fixed by an equality (=) condition in the WHERE clause.

Database: {database}

Generated Clauses: {generated_clauses}

Return the results in a FLAT JSON format.
**DO NOT** include any explanations or notes in the output. **ONLY** return JSON.

## HAVING Clause Generation

You are both an NBA fan and an SQL expert. Given the provided database and the generated clauses, generate a HAVING clause that reflects authentic NBA-related curiosity.

Ensure the following requirements:

- Output Structure: Return a JSON object containing a single key, `"having"`, with its value being a HAVING clause.
- Ensure NBA Fan Relevance: Generate the HAVING clause that aligns naturally with realistic and meaningful queries that NBA fans are likely to ask.
- Maintain Specificity and Clarity of Intent: Generate a well-defined and clear HAVING clause without making it overly narrow or contrived.
- Align with Generated Clauses: Ensure that the HAVING clause maintains logical consistency with previously generated clauses, preserving semantic coherence.
- Ensure Synthetic Correctness: Generate the HAVING clause that is syntactically correct and executable on the provided database.

Database: `{database}`

Generated Clauses: `{generated_clauses}`

Return the results in a FLAT JSON format.
**DO NOT** include any explanations or notes in the output. **ONLY** return JSON.

## ORDER BY Clause Generation

You are both an NBA fan and an SQL expert. Given the provided database and the generated clauses, generate an ORDER BY clause that reflects authentic NBA-related curiosity.

Ensure the following requirements:

- Output Structure: Return a JSON object containing a single key, `"order"`, with its value being an ORDER BY clause.
- Ensure NBA Fan Relevance: Generate the ORDER BY clause that aligns naturally with realistic and meaningful queries that NBA fans are likely to ask.
- Align with Generated Clauses: Ensure that the ORDER BY clause maintains logical consistency with previously generated clauses, preserving semantic coherence.
- Ensure Synthetic Correctness: Generate the ORDER BY clause that is syntactically correct and executable on the provided database.

Database: `{database}`

Generated Clauses: `{generated_clauses}`

Return the results in a FLAT JSON format.
**DO NOT** include any explanations or notes in the output. **ONLY** return JSON.

**LIMIT Clause Generation**

You are both an NBA fan and an SQL expert. Given the provided database and the generated clauses, generate a LIMIT clause that reflects authentic NBA-related curiosity.

Ensure the following requirements:

- Output Structure: Return a JSON object containing a single key, `"limit"`, with its value being a LIMIT clause.
- Ensure NBA Fan Relevance: Generate the LIMIT clause that aligns naturally with realistic and meaningful queries that NBA fans are likely to ask.
- Align with Generated Clauses: Ensure that the LIMIT clause maintains logical consistency with previously generated clauses, preserving semantic coherence.
- Ensure Synthetic Correctness: Generate the LIMIT clause that is syntactically correct and executable on the provided database.

Database: `{database}`

Generated Clauses: `{generated_clauses}`

Return the results in a FLAT JSON format.
**DO NOT** include any explanations or notes in the output. **ONLY** return JSON.

---

**SELECT Clause (Non-Aggregate)**

You are both an NBA fan and an SQL expert. Given the provided database and the generated clauses, generate a SELECT clause that specifies a necessary field for retrieving meaningful NBA-related data.

Ensure the following requirements:

- Output Structure: Return a JSON object containing a single key, `"select"`, with its value being a SELECT clause that projects a single column without an aggregation function meaningfully.
- Ensure NBA Fan Relevance: Generate the SELECT clause that aligns naturally with realistic and meaningful queries that NBA fans are likely to ask.
- Align with Generated Clauses: Ensure that the SELECT clause maintains logical consistency with previously generated clauses, preserving semantic coherence.
- Ensure Synthetic Correctness: Generate the SELECT clause that is syntactically correct and executable on the provided database.

**IMPORTANT**: Do not project columns used in the WHERE clause.

Database: `{database}`

Generated Clauses: `{generated_clauses}`

Return the results in a FLAT JSON format.
**DO NOT** include any explanations or notes in the output. **ONLY** return JSON.

---

**SELECT Clause (Aggregate)**

You are both an NBA fan and an SQL expert. Given the provided database and the generated clauses, generate a SELECT clause that aggregates a single column for retrieving meaningful NBA-related statistics.

Ensure the following requirements:

- Output Structure: Return a JSON object containing a single key, `"select"`, with its value being a SELECT clause that aggregates (MAX, MIN, AVG, or COUNT, etc.) a single column meaningfully.
- Ensure NBA Fan Relevance: Generate the SELECT clause that aligns naturally with realistic and meaningful queries that NBA fans are likely to ask.
- Align with Generated Clauses: Ensure that the SELECT clause maintains logical consistency with previously generated clauses, preserving semantic coherence.
- Ensure Synthetic Correctness: Generate the SELECT clause that is syntactically correct and executable on the provided database.

Database: `{database}`

Generated Clauses: `{generated_clauses}`

Return the results in a FLAT JSON format.
**DO NOT** include any explanations or notes in the output. **ONLY** return JSON.

**Inner Query Block Selection**

You are both an NBA fan and an SQL expert. Given the provided database, generated clauses, and the candidate inner query blocks, select the most appropriate inner query block for generating a nested predicate that reflects authentic NBA-related curiosity.

Select the most appropriate inner query block to generate a nested predicate that aligns naturally with realistic and meaningful **multi-hop** queries NBA fans are likely to ask.

Your output must be in JSON format with the key:

- `"inner_query_block"`: Select the most appropriate inner query block from the Candidate Inner Query Blocks.

**IMPORTANT**:

- Do not select the inner query block that has already been used in the generated clauses and is not included in the candidate inner query blocks.

Database: `{schema}`

Generated FROM Clause: `{generated_from_clause}`

Generated WHERE Clause: `{generated_where_clause}`

Candidate Inner Query Blocks: `{candidate_inner_query_blocks}`

Return the results in a FLAT JSON format.
**DO NOT** include any explanations or notes in the output. **ONLY** return JSON.

### FROM Clause Generation

You are both an NBA fan and an SQL expert. Given the database and the inner query block, generate a **FROM** clause of the outer query block that reflects authentic NBA-related curiosity.

Ensure the following requirements:

- **Output Structure**: Return a JSON object containing a single key, `"from"`, with its value being **a single-table** FROM clause of the outer query block from the provided database (i.e., do not include any sub-selects or nested queries directly in the FROM clause).
- **Ensure NBA Fan Relevance**: Generate the FROM clause that aligns naturally with realistic and meaningful **multi-hop** queries that NBA fans are likely to ask.
- **Ensure Synthetic Correctness**: Generate the FROM clause that is syntactically correct and executable on the provided database.
- **Separate Inner Query**: The inner query block must remain separate; it should later be incorporated into the WHERE clause, not nested in the FROM clause.
- **Ensure Natural Connection**: Choose an outer table whose columns can be naturally referenced or filtered against the results of the inner query block.

**IMPORTANT**: If the inner query block performs aggregation in the SELECT clause and no outer table includes the aggregated columns, reuse the table referenced in the inner query as the outer table.

Database: `{schema}`

Inner Query Block: `{subquery}`

Return the results in a FLAT JSON format.
**DO NOT** include any explanations or notes in the output. **ONLY** return JSON.

**Type-N Nested Predicate Generation**

You are both an NBA fan and an SQL expert. Based on the given database, generated clauses, selected inner query block Q, and its execution result, generate a **type-n** nested predicate that reflects authentic NBA-related curiosity.

Ensure the following requirements:

- Ensure **type-n** Nesting: The inner query block Q must **not** contain a join predicate that references the relation of the outer query block, and its SELECT clause must project a column without an aggregate function.
- Ensure NBA Fan Relevance: Generate the nested predicate that aligns naturally with realistic and meaningful **multi-hop** queries that NBA fans are likely to ask.
- Ensure Synthetic Correctness: Generate the nested predicate that is syntactically correct and executable on the provided database.
- Ensure Semantic Alignment: If the inner query's SELECT column does not semantically match any column in the outer query's table, revise it for consistency.

The **type-n** nested predicate must be in the form: `OuterTable.column [IN | NOT IN] ( Q )`.

Your output must be in JSON format with the keys:

- `"nested_predicate"`: Only the type-n nested predicate based on the selected inner query block.
- `"logical_operator"`: If a WHERE clause exists, return 'AND' or 'OR'.

**IMPORTANT**:

- Ensure that the nesting level of the inner query block is correctly preserved. The expected nesting level is `{height}`.
- Do not modify the nesting level of the provided inner query block.

Database: `{schema}`

Generated FROM Clause of the Outer Query: `{generated_from_clause}`

Generated WHERE Clause of the Outer Query: `{generated_where_clause}`

Selected Inner Query Block Q: `{selected_inner_query_block}`

Return the results in a FLAT JSON format.
**DO NOT** include any explanations or notes in the output. **ONLY** return JSON.

---

**Type-A Nested Predicate Generation**

You are both an NBA fan and an SQL expert. Based on the given database, generated clauses, selected inner query block, and its execution result, generate a **type-a** nested predicate that reflects authentic NBA-related curiosity.

Ensure the following requirements:

- Ensure **type-a** Nesting: The inner query block Q must not contain a join predicate referencing the outer query's relation, and its SELECT clause must contain an aggregate function associated with a column.
- Ensure NBA Fan Relevance: Generate the nested predicate that aligns naturally with realistic and meaningful **multi-hop** queries that NBA fans are likely to ask.
- Ensure Synthetic Correctness: The predicate must be executable and logically valid over the schema.

The **type-a** nested predicate must follow the form:
```
OuterTable.column [= | != | < | <= | > | >=] ( Q with
aggregate function )
```

Your output must be in JSON format with the keys:

- `"nested_predicate"`: Only the type-a nested predicate based on the selected inner query block.
- `"logical_operator"`: If a WHERE clause exists, return 'AND' or 'OR'.

**IMPORTANT**:

- Do not revise the SELECT clause of the Q.
- Ensure that the nesting level remains `{height}`.

Database: `{schema}`

Generated FROM Clause of the Outer Query: `{generated_from_clause}`

Generated WHERE Clause of the Outer Query: `{generated_where_clause}`

Selected Inner Query Block Q: `{selected_inner_query_block}`

Return the results in a FLAT JSON format.
**DO NOT** include any explanations or notes in the output. **ONLY** return JSON.

**Type-J Nested Predicate Generation**

You are both an NBA fan and an SQL expert. Based on the given database, generated clauses, selected inner query block, and its execution result, generate a **type-j** nested predicate that reflects authentic NBA-related curiosity.

Ensure the following requirements:

- Ensure **type-j** Nesting: Revise the inner query block Q to ensure it includes a join predicate in its WHERE clause that references the outer query's relation, and its SELECT clause must project a column without an aggregate function.
- Ensure NBA Fan Relevance: Generate the nested predicate that aligns naturally with realistic and meaningful **multi-hop** queries that NBA fans are likely to ask.
- Ensure Synthetic Correctness: Generate the nested predicate that is syntactically correct and executable on the provided database.
- Ensure Semantic Alignment: If the inner query's SELECT column does not semantically match any column in the outer query's table, revise it for consistency.

The **type-j** nested predicate must be in one of the following forms:
```
OuterTable.column [IN | NOT IN] (SELECT ...  FROM ...  WHERE
...  [join predicate] ...)
```
or
```
[EXISTS | NOT EXISTS] (SELECT ...  FROM ...  WHERE ...
[join predicate] ...)
```

Your output must be in JSON format with the keys:

- `"nested_predicate"`: Only the type-j nested predicate based on the selected inner query block.
- `"logical_operator"`: If a WHERE clause exists, return 'AND' or 'OR'.

**IMPORTANT**:

- The join predicate involving the outer query's relation must appear in the **WHERE clause** of Q, not its FROM clause.
- The expected nesting level is {height}. Do not modify it.

Database: {schema}

Generated FROM Clause of the Outer Query: {generated_from_clause}

Generated WHERE Clause of the Outer Query: {generated_where_clause}

Selected Inner Query Block Q: {selected_inner_query_block}

Return the results in a FLAT JSON format.
**DO NOT** include any explanations or notes in the output. **ONLY** return JSON.

**Type-JA Nested Predicate Generation**

You are both an NBA fan and an SQL expert. Based on the given database, generated clauses, selected inner query block, and its execution result, generate a **type-ja** nested predicate that reflects authentic NBA-related curiosity.

Ensure the following requirements:

- Ensure **type-ja** Nesting: Revise the inner query block Q to include a join predicate in its WHERE clause that references the outer query's relation and ensure its SELECT clause contains an aggregate function.
- Ensure NBA Fan Relevance: Generate the nested predicate that aligns naturally with realistic and meaningful **multi-hop** queries that NBA fans are likely to ask.
- Ensure Synthetic Correctness: The resulting predicate must be executable and valid over the database schema.

The **type-ja** nested predicate must follow one of the forms:
```
OuterTable.column [= | != | < | <= | > | >=] (SELECT [agg]
...  FROM ...  WHERE ...  [join predicate] ...)
```
or
```
[EXISTS | NOT EXISTS] (SELECT [agg] ...  FROM ...  WHERE ...
[join predicate] ...)
```

Your output must be in JSON format with the keys:

- `"nested_predicate"`: Only the type-ja nested predicate based on the selected inner query block.
- `"logical_operator"`: If a WHERE clause exists, return 'AND' or 'OR'.

**IMPORTANT**:

- The join predicate involving the outer query's relation must appear in the **WHERE clause**, not the FROM clause.
- Do not revise the SELECT clause of the Q.
- Do not modify the nesting level (`{height}`).

Database: `{schema}`

Generated FROM Clause of the Outer Query: `{generated_from_clause}`

Generated WHERE Clause of the Outer Query: `{generated_where_clause}`

Selected Inner Query Block Q: `{selected_inner_query_block}`

Return the results in a FLAT JSON format.
**DO NOT** include any explanations or notes in the output. **ONLY** return JSON.

**Provenance-based Refinement**

You are both an NBA fan and an SQL expert. Based on the given original SQL query, provenance analysis results, and problematic subquery or condition which filters out all the rows, fix the original query's problematic subquery or condition so that it retrieves some results from the database.

Ensure the following requirements:
1) **Output Structure**: Return a JSON object containing a single key, `"corrected_query"`, with its value being the corrected SQL query.
2) **Ensure NBA Fan Relevance**: Maintain the original query's NBA-related curiosity and focus on realistic and meaningful queries that NBA fans are likely to ask.

**IMPORTANT**:

- You may add an additional predicate in the inner query or adjust the filtering threshold within the problematic subquery Q to intentionally include the important rows or exclude outlier rows (e.g., those with extremely high or low values) that overly constrain the outer query.

- You may also adjust the comparison operator (e.g., > to >=, < to <=) or the value of the problematic condition to relax the filtering criteria.

- Do **not** delete the join predicate in the WHERE clause of the problematic subquery Q (e.g., `WHERE outer_table_name.column_name = inner_table_name.column_name`).

**Original SQL Query**: `{query}`
**Problematic Condition**: `{problematic_condition}`
**Problematic Subquery Q**: `{problematic_subquery}`
**Execution Result of the Subquery Q**: `{problematic_subquery_execution _result}`
**Provenance Analysis Results**: `{provenance_analysis_results}`

Return the results in a **FLAT JSON**.
**NEVER include ANY EXPLANATION or NOTE in the output, ONLY OUTPUT JSON.**

**SQL Query Naturalness Evaluation**

You have a set of evaluation criteria to judge whether a given SQL query reflects a question that is likely to be asked by a typical person.

When evaluating the query, refer to the following points:

1. **Relevance**:
   - **Definition**: Measures how likely it is that a real person would be interested in the query.
   - **Low Score (1)**: The query covers obscure or highly technical aspects unrelated to typical person discussions (e.g., internal database IDs or rarely discussed statistics).
   - **High Score (5)**: The query reflects a common, popular interest among people (e.g., game stats, player/team information, draft results, etc.).

2. **Specificity & Clarity of Intent**:
   - **Definition**: Evaluates whether the question is clearly targeted and sufficiently detailed to reveal a genuine NBA-related interest—without being so narrow as to be contrived.
   - **Low Score (1)**: The query is too vague ("Show me some NBA data") or overly convoluted/contrived.
   - **High Score (5)**: The query clearly captures a plausible question (e.g., "Which NBA player scored the most points in home games last month?").

3. **Overall Naturalness**:
   - Combine the above criteria and decide if the query is "natural" (likely to be asked by a real person) or "unnatural".
   - The query is considered natural if its overall score is 3 or higher.

Your output must be in JSON format with the following keys:

- `"relevance_score"`: Integer from 1 to 5.
- `"specificity_clarity_of_intent_score"`: Integer from 1 to 5.
- `"overall_naturalness_score"`: Integer from 1 to 5.
- `"reason"`: Explanation referencing the scores and justifying whether the query is considered natural or unnatural.

Database Schema: `{database_schema}`

SQL Query Template: `{question}`

Return the results in a FLAT JSON format.
**NEVER include ANY EXPLANATION or NOTE in the output, ONLY OUTPUT JSON.**

