# OpenReview forum: "SPARTA: Scalable and Principled Benchmark of Tree-Structured Multi-hop QA over Text and Tables"
_ICLR.cc/2026/Conference — ICLR 2026 Poster_

### Official Review · Reviewer_N2ch · 2025-10-29

**Soundness:** 2
**Presentation:** 2
**Contribution:** 2
**Rating:** 4
**Confidence:** 4

**Summary:**

SPARTA introduces a large-scale, principled benchmark for Table-Text Question Answering (QA) by addressing critical limitations—shallow reasoning, small data scale, and annotation noise—inherent in prior manual datasets like HybridQA. The core methodology involves constructing a unified Reference Fact Database by incorporating text-derived atomic facts into structured Grounding Tables, enabling an LLM-guided pipeline to synthesize highly complex, executable SQL queries that model multi-hop, tree-structured reasoning across modalities. Utilizing novel techniques like provenance-based refinement and post-order structural enforcement, SPARTA efficiently generates thousands of high-fidelity question-answer pairs covering advanced analytical operations such as grouping and aggregation, leading to a profound performance collapse (over 30 F1 points) in state-of-the-art QA models, thereby establishing a new, rigorous standard for cross-modal reasoning evaluation.

**Strengths:**

Strength 1: High-Fidelity Scaling and Data Rigor SPARTA dramatically increases the realism of Table-Text QA by operating over large-scale databases featuring thousands of rows (e.g., 3,280.5 mean cardinality), moving well beyond the small, toy-scale web tables typically used in previous benchmarks. This substantial increase in data size rigorously evaluates model performance in environments demanding true large-database reasoning and efficient indexing.


Strength 2: Comprehensive Coverage of Tree-Structured Multi-Hop and Analytical Reasoning The framework systematically generates questions covering advanced analytical operations such as aggregation and grouping, which were largely absent from prior benchmarks. Furthermore, it features deep, tree-structured multi-hop reasoning, which goes beyond the simplistic linear chains of inference found in existing QA resources.


Strength 3: Cost-Effective Rigor via Provenance The generation pipeline achieves high data quality and efficiency by employing Provenance-based Refinement, guaranteeing every synthesized SQL query is executable and returns a non-empty result. This mechanism ensures semantic validity while drastically reducing the cost of generation, requiring only one quarter of the annotation time compared to manual benchmarks like HybridQA.

**Weaknesses:**

Weakness 1: Undocumented Quality of Atomic Fact Extraction The crucial first step of automatically extracting atomic facts from unstructured text into grounding tables lacks reported validation or error analysis. If errors in this unquantified automated extraction process propagate, they could undermine the benchmark's fundamental data grounding and rigor.


Weakness 2: Acute Cross-Modal Fragility in Program-Based Models The inclusion of unstructured text causes a significant F1 score drop (e.g., 63.9% for HProPro), indicating fundamental brittleness in program-based reasoning when integrating facts from relational tables and textual grounding tables. This massive performance collapse suggests inadequate logical bridging and factual integration capabilities, especially for complex analytical queries that span the heterogeneous data store.


Weakness 3: Vulnerability to Correlated Joins and Aggregation Models demonstrate pronounced vulnerability to the synergistic complexity introduced by combining correlated joins and aggregation (Type-JA queries), yielding the lowest reported F1 scores. This quantitative failure highlights that state-of-the-art models struggle with highly constrained, context-sensitive analytical calculations typical of advanced database logic.


Weakness 4: Scaling Limitations Across Query Structural Dimensions Empirical results show model performance degrades consistently and sharply as query complexity increases along either the depth (chain length) or breadth (branching structure) dimensions of the query graph. This suggests current methods fail to generalize tree-structured reasoning, exhibiting fundamental computational limitations in managing multiple, concurrent relational paths.


Weakness 5: Costly Dependence on LLM Corrective Feedback Despite efficiency efforts, the nested query generation pipeline is inherently reliant on numerous expensive error-correction loops, requiring 53.6% more LLM calls than the theoretical ideal. This high overhead dedicated solely to provenance-based repair reveals the intrinsic unreliability of LLMs in autonomously generating semantically sound SQL without database execution feedback.


Weakness 6: Limited Scope and Insufficient Validation Granularity The benchmark is confined strictly to the Table-Text modality, neglecting other real-world data sources like images or video, thus limiting its applicability in the evolving multimodal QA landscape. Furthermore, the human validation is explicitly "lightweight," avoiding verification of the complex multi-hop reasoning path itself, which risks overlooking subtle semantic misalignment between the generated SQL and the final natural language question.

**Questions:**

same as weakness

---

> ### Author Response · Authors · 2025-11-19
> **Response to Reviewer N2ch (1/2)**
>
> Dear Reviewer N2ch,
>
> Thank you for your valuable comments. We have carefully addressed your suggestion. Please find below our detailed responses. We believe these revisions, guided by your comments, will greatly improve the clarity and readability of our work. Please let us know if there is anything further we need to address to help you consider updating your score.
>
> ---
>
> **Response to W1**:
> Thank you for your valuable question. We would like to clarify that the atomic fact extraction process briefly described in Section 1 is only one of two strategies we employ to construct the grounding tables introduced in Section 3.2.  Our benchmark consists of three datasets, and each of them is constructed via a different strategy with its own quality guarantees. We clarify below how atomic fact quality is controlled in each case.
> 1) ROTOWIRE: For the NBA domain, we use the ROTOWIRE dataset as part of our reference fact database, whose structured tables are widely used as gold supervision for text-to-table and have been verified by the authors of [1] for consistency with the accompanying game reports.
> 2) The other two datasets: For the remaining two datasets, we do not perform text-to-table extraction from raw text. Instead, as described in Section 3.2, we adopt a table-to-text–based strategy: we start from existing structured tables and convert a subset of them (3 out of 6 for Movie domain and 2 out of 5 for Finance domain) into grounding tables using rule-based templates. This transformation is deterministic and template-driven, so the atomic facts are generated without model-induced extraction errors. We manually design and check these templates, which prevents spurious facts from being introduced in these two datasets.
>
> To enhance clarity, we have revised Section 1 to explicitly distinguish these dataset-specific construction strategies and to point readers to Section 3.2, where both strategies and their reliability are discussed in more detail.
>
> [1] Xueqing Wu, Jiacheng Zhang, and Hang Li. Text-to-table: A new way of information extraction. In Proceedings of the 60th Annual Meeting of the Association for Computational Linguistics (Volume 1: Long Papers), pp. 2518–2533, 2022.
>
> ---
>
> **Response to W2:**
> We would like to clarify that the low performance of the program-based model is not due to inadequate logical bridging in our benchmark. Instead, it deliberately exposes the models' inherent weaknesses in cross-modal factual integration.
>
> The queries in SPARTA are systematically constructed to ensure that the logical bridges are natural and well-formed. This is achieved by generating queries based on explicit join relationships within the underlying reference fact databases:
> - For the Medical and Movie domains, we utilized publicly available relational databases. The established foreign-key and join relationships in these databases guarantee coherent logical connections between tables.
> - For the NBA domain, we carefully established join relationships between the ROTOWIRE dataset and other public, structured datasets, ensuring consistent linking between textual grounding tables and relational tables.
>
> To verify consistency, we analyzed the results from Table 6 broken down by domain. We observe a consistent performance drop across all three domains, indicating that the observed degradation is a systemic challenge for current program-based models in cross-modal settings, rather than a deficiency of the benchmark itself.
>
> | Benchmark        | Setting                     | F1 (HProPro w/ GPT-5) |
> |------------------|------------------------------|------------------------|
> | **SPARTA (NBA)** | Table-Only Reasoning         | 49.7                   |
> |                  | Table–Text Cross Reasoning   | 10.2 (-79.5%)          |
> | **SPARTA (Medical)** | Table-Only Reasoning     | 46.4                   |
> |                  | Table–Text Cross Reasoning   | 15.9 (-65.7%)          |
> | **SPARTA (Movie)** | Table-Only Reasoning       | 50.3                   |
> |                  | Table–Text Cross Reasoning   | 35.5 (-29.5%)          |
>
> ---

---

> ### Author Response · Authors · 2025-11-19
> **Response to Reviewer N2ch (2/2)**
>
> **Response to W3 & W4:**
> We would like to clarify that our primary contribution is the SPARTA framework and the resulting benchmarks, rather than a new table-text QA method. Therefore, the vulnerabilities you identified such as the low F1 scores on Type-JA queries (W3) and the performance degradation as query depth and breadth increase (W4), are not shortcomings of our work but  key findings uncovered by our benchmarks. Indeed, multiple reviewers (e.g., JYtH Strength 3, kiAT Strength 3, iHeT Strength 3) explicitly recognized that exposing these limitations is a major strength of our work. To avoid any misinterpretation, we have revised Section 4.4 to make our positioning more explicit, clearly stating that these observed model failures are intended outcomes of our analysis and underscore the value of SPARTA in identifying critical directions for future research on complex, table-text query reasoning.
>
> ---
>
> **Response to W5:**
> We agree that current LLMs are not reliable enough to generate semantically correct SQL without feedback, and our design explicitly embraces this reality: provenance-based refinement is introduced precisely to control the cost of such corrective loops rather than to eliminate them.
>
> To demonstrate that this refinement remains effective and cost-efficient regardless of the LLM’s inherent reliability, we conducted additional experiments across LLMs with different capacities. In addition to Llama-3.1-70B-Instruct, we evaluated a smaller LLM (gpt-oss-20B) and a larger LLM (gpt-oss-120B). Across all three models, Post-Order + Prov is the most cost-effective strategy, requiring 4,854, 4,722, and 3,831 calls, respectively, while reducing call volume by 18.8%, 42.8%, and 54.5% relative to vanilla Post-Order, and by 64.7%, 66.2%, and 65.9% relative to One-Shot-k.
>
> These results show that, although some overhead above the theoretical “Ideal Calls” is unavoidable with today’s LLMs, disciplined post-order construction combined with provenance-driven repair consistently minimizes redundant generations. We have added these results and the precise definition of the “Ideal Calls” metric to Appendix F of the revised manuscript.
>
> | LLM                    | Method                     | Ideal Calls | Total Calls |
> |------------------------|----------------------------|-------------|-------------|
> | **gpt-oss-20B**        | One-Shot-k                 | 2661        | 13736       |
> |                        | Post-Order (no provenance) | 3073        | 5977        |
> |                        | Post-Order + Prov          | 3146        | 4854        |
> | **Llama-3.1-70B-Instruct** | One-Shot-k             | 2664        | 13962       |
> |                        | Post-Order (no provenance) | 3104        | 8253        |
> |                        | Post-Order + Prov          | 3074        | 4722        |
> | **gpt-oss-120B**       | One-Shot-k                 | 2621        | 11225       |
> |                        | Post-Order (no provenance) | 3152        | 8425        |
> |                        | Post-Order + Prov          | 3145        | 3831        |
>
> ---
>
> **Response to W6:**
> We agree that extending SPARTA beyond table–text to additional modalities such as images or videos is an important direction. This limitation and its multimodal extension were already discussed in Appendix H; in the revised manuscript, we have moved this discussion into the main body to make the scope and future directions more explicit.
>
> Regarding “lightweight” validation, this term refers to efficiency rather than a relaxation of rigor. Our annotators validate natural language questions against existing SQL queries whose semantics are already grounded in the database, rather than constructing queries and reasoning paths from scratch. Unlike HybridQA where annotators must read a table plus on average 33.5 passages to manually discover multi-hop chains our validators primarily inspect table schemas and explicit join relationships over a small number of tables (typically just a few per query), which makes the check very concise while still ensuring semantic alignment between the question and the SQL. As reported in Section 3.4, this design reduces per-query annotation time to about one quarter of HybridQA for the same number of queries, despite SPARTA generating substantially more diverse and structurally complex queries.

---

> > ### Comment · Reviewer_N2ch · 2025-11-20
> >
> > Thank you for the reply and the rebuttal solved my issues, I chaged the rating to 6

---

> ### Author Response · Authors · 2025-11-20
> **Sincere Gratitude from Authors**
>
> Thank you for taking the time to review our responses and revising your evaluation. We are delighted to hear that we have addressed your concerns. Your insightful feedback has been instrumental in improving the quality of our work.

---

### Official Review · Reviewer_iHeT · 2025-10-31

**Soundness:** 2
**Presentation:** 3
**Contribution:** 2
**Rating:** 4
**Confidence:** 3

**Summary:**

This paper introduces SPARTA, a large-scale Table–Text Question Answering (QA) benchmarks. Unlike existing small and shallow datasets, SPARTA generates complex multi-hop and aggregation-based QA pairs by synthesizing executable SQL queries and corresponding natural-language questions. Through provenance-based refinement and realistic-structure enforcement, it produces thousands of high-quality question–answer pairs with only a quarter of HybridQA’s annotation cost, revealing significant performance drops in state-of-the-art models and highlighting current limitations in cross-modal reasoning.

**Strengths:**

1. The paper is clear and easy to follow.
2. The analysis of existing benmarks is comprehensive.
3. Several experiments along with analysis on this benchmark.

**Weaknesses:**

1. **Excessive Dependence on High-Capacity LLMs for Pipeline Efficiency**.
The efficiency of the Provenance-based Refinement loop relies heavily on the advanced reasoning capability of a large LLM (Llama-3.1-70B-Instruct) to accurately diagnose and correct erroneous SQL predicates. However, the paper does not analyze how the framework’s quality and cost-effectiveness (central to its “Scalable” claim) would be affected when using smaller or less capable LLMs. This raises concerns about the robustness, reproducibility, and general accessibility of the proposed generation pipeline.

2. **Lack of Fine-Grained Error Attribution in End-to-End Settings.**
The error analysis highlights major failure categories such as “Relevant data missing” and “Erroneous data analysis”. However, in the end-to-end setting, it remains unclear whether these failures stem from inherent weaknesses in the reasoning and program execution components or from compounded upstream retrieval errors. A more fine-grained breakdown of error contributions across the Retrieval and Reasoning stages is essential to accurately diagnose the true limitations revealed by SPARTA.

3. **Limited Rigor in Query Naturalness Evaluation.**
The evaluation of question naturalness relies on verbalizing SQL queries using LLMs (AST-ICL and ChatGPT-4o). This top-down SQL → NL generation process may not faithfully reflect the organic intent or linguistic variability of real user questions. As a result, the claimed “realism” of the complex, tree-structured questions could be biased or overstated.

4. **Oversimplification of Reference Fact Database Construction.** The foundational step of the pipeline involves merging source tables with Grounding Tables, which are created by automatically extracting atomic facts from unstructured text passages. However, the paper primarily relies on a pre-processed corpus (ROTOWIRE) within the NBA domain, and provides insufficient technical detail on the general and robust mechanisms for extracting and normalizing atomic facts from arbitrary text into structured Grounding Tables. This omission raises concerns about the reliability and domain-agnostic portability of the proposed pipeline.

5. This dataset contains three subsets only, which could be expanded in to larger range of QAs.


Overall, I believe there is still considerable room for improvement in this paper, and the weaknesses outweigh the strengths. Therefore, I assign a final rating of borderline reject.

**Questions:**

Please see weaknesses.

---

> ### Author Response · Authors · 2025-11-19
> **Response to Reviewer iHeT (1/2)**
>
> Dear Reviewer iHeT,
>
> Thank you for your valuable comments and insightful suggestions. We have carefully addressed each point and performed additional analyses. Please find our detailed responses below. We believe these revisions, guided by your feedback, significantly enhance the strength and clarity of our proposed method. Please let us know if there is anything further we need to address to help you consider updating your score.
>
> ---
>
> **Response to W1**:
> Thank you for your thoughtful comments. To demonstrate that our provenance-based refinement is effective regardless of the LLM’s size, we conducted additional experiments comparing generation cost across LLMs with different capacities (i.e., parameter sizes). Specifically, in addition to the Llama-3.1-70B-Instruct model evaluated in the manuscript, we measured generation cost using a smaller-parameter LLM (gpt-oss-20B) and a larger-parameter LLM (gpt-oss-120B).
>
> | LLM                    | Method                     | Ideal Calls | Total Calls |
> |------------------------|----------------------------|-------------|-------------|
> | **gpt-oss-20B**        | One-Shot-k                 | 2661        | 13736       |
> |                        | Post-Order (no provenance) | 3073        | 5977        |
> |                        | Post-Order + Prov          | 3146        | 4854        |
> | **Llama-3.1-70B-Instruct** | One-Shot-k             | 2664        | 13962       |
> |                        | Post-Order (no provenance) | 3104        | 8253        |
> |                        | Post-Order + Prov          | 3074        | 4722        |
> | **gpt-oss-120B**       | One-Shot-k                 | 2621        | 11225       |
> |                        | Post-Order (no provenance) | 3152        | 8425        |
> |                        | Post-Order + Prov          | 3145        | 3831        |
>
> Our experimental results show that Post-Order + Prov is the most cost-effective approach across all three LLM variants, completing with 4854, 4722, and 3831 calls for the respective models, while reducing call volume by 18.8%, 42.8%, and 54.5% relative to vanilla Post-Order, and by 64.7%, 66.2%, and 65.9% relative to One-Shot-k. These results indicate that disciplined post-order construction combined with provenance-driven repair consistently minimizes redundant generations independent of the LLM's scale. We have incorporated these new results into Appendix F of the revised manuscript to further support the robustness of our method.
>
> ---
>
> **Response to W2**:
> To address the concern, we conducted an additional error analysis focusing on questions where HProPro answered incorrectly (F1=0) in the end-to-end setting. Our results show that 39.2% of total errors originate from the retrieval stage, while the remaining 60.8% arise from the reasoning stage. This quantitative breakdown confirms that the primary bottleneck lies in the model's reasoning.
>
> For a more detailed breakdown of the reasoning stage errors, we sampled 100 queries and manually annotated their error types. The resulting distribution closely mirrored our prior analysis in the SPARTA (Oracle) setting, with "Relevant Data Missing" comprising 38% and "Erroneous Data Analysis" comprising 42% as the major failure categories, indicating that the dominant failure modes are consistent between the Oracle and end-to-end settings. We have incorporated this detailed breakdown into Appendix J of a revised paper for a clearer diagnosis of model limitations in end-to-end settings.
>
> ---
>
> **Response to W3**:
> We would like to clarify that the naturalness evaluation is conducted on the “SQL queries,” rather than directly on their NL verbalizations. The translated questions and their query results subsequently undergo final validation by human evaluators, who filter out semantically odd or unrealistic cases, ensuring that only realistic queries are retained in the benchmark. We did not intend this benchmark to capture linguistic variability of user questions, as this dimension can be evaluated separately in other benchmarks such as [1]. To address any ambiguity, we have revised the description of the measures in Section 4.2 based on the details provided in Appendix N.
>
>
> [1] Gan, Yujian, et al. "Towards Robustness of Text-to-SQL Models against Synonym Substitution." Proceedings of the 59th Annual Meeting of the Association for Computational Linguistics and the 11th International Joint Conference on Natural Language Processing (Volume 1: Long Papers). 2021.

---

> ### Author Response · Authors · 2025-11-19
> **Response to Reviewer iHeT (2/2)**
>
> **Response to W4:**
> We would like to clarify that Section 3.2 outlines two complementary strategies for building Grounding Tables: a text-to-table approach and an alternative table-to-text approach. In our work, we primarily leverage the table-to-text-derived Grounding tables to demonstrate the domain-agnostic portability of the proposed pipeline. This method employs rule-based techniques, yielding deterministic table construction and 100% high-precision facts.
>
> For text-to-table scenarios, numerous text-to-table methods [2, 3, 4] exist. To ensure highly accurate facts, these methods typically require final human validation. The ROTOWIRE corpus is one such example: its tables and textual descriptions have been validated by the authors of [2], providing a high-quality testbed for our pipeline. Developing a 100%-accurate text-to-table extractor is an orthogonal research direction and not our primary contribution; rather, our work focuses on effectively utilizing such text–table pairs once they are available within our pipeline.
>
> [2] Xueqing Wu, Jiacheng Zhang, and Hang Li. Text-to-table: A new way of information extraction. In Proceedings of the 60th Annual Meeting of the Association for Computational Linguistics (Volume 1: Long Papers), pp. 2518–2533, 2022.
>
> [3] Naman Ahuja, Fenil Bardoliya, Chitta Baral, and Vivek Gupta. 2025. Map&Make: Schema Guided Text to Table Generation. In Proceedings of the 63rd Annual Meeting of the Association for Computational Linguistics (Volume 1: Long Papers), pages 30249–30262, Vienna, Austria. Association for Computational Linguistics.
>
> [4] Harne, Sarthak, and Arvind Agarwal. "Llm driven text-to-table generation through sub-tasks guidance and iterative refinement." arXiv preprint arXiv:2508.08653 (2025).
>
> ---
>
> **Response to W5:**
> Thank you for your suggestion to expand the dataset. We limited the benchmarks to three domains because constructing each subset required human verification for the final natural language query-response pairs. To facilitate broader adoption and future expansion to additional domains, we will make our entire data generation framework and code publicly available.

---

> ### Author Response · Authors · 2025-11-27
> **A Gentle Reminder**
>
> Dear Reviewer iHeT,
>
> Thank you again for your thoughtful and constructive comments, which have been helpful for improving our submission.
>
> As the author–reviewer discussion period is coming to an end, we would like to kindly confirm that you have had a chance to review our detailed rebuttal. We believe that the main technical concerns you raised (concerns on robustness with varying LLM sizes, lack of fine-grained error attribution in end-to-end settings, limited rigor in query naturalness evaluation, insufficient technical detail on grounding tables, suggestion for dataset expansion) are now thoroughly addressed in our response and revisions.
>
> Additionally, because these updates and clarifications were made specifically in response to your constructive feedback, we kindly hope that they will encourage you to reconsider your evaluation. Should you have any remaining questions or reservations, please feel free to let us know at any time—we are fully committed to addressing all concerns to your satisfaction.
>
> Best,
>
> Authors

---

> ### Comment · Reviewer_iHeT · 2025-11-27
>
> Thank you for detailed responses, which address most of my concerns. I decide to raise my score.

---

> > ### Author Response · Authors · 2025-11-28
> > **Sincere Gratitude from Authors**
> >
> > Thank you for your response. We are pleased to hear that our reply has addressed your concerns. Your insightful suggestions have been instrumental in enhancing the quality of our paper.
> >
> > We noticed that the score does not yet appear to be updated in the system. If appropriate, could you kindly check this at your convenience?

---

> > > ### Comment · Reviewer_iHeT · 2025-11-28
> > >
> > > I found that the 'Edit' option is no longer available in the system, so I was unable to change my score. I will update my score accordingly once I figure out what happened.

---

> > > > ### Author Response · Authors · 2025-11-28
> > > >
> > > > Thank you for letting us know. We will look forward to your updated score once the system issue is resolved.

---

### Official Review · Reviewer_kiAT · 2025-11-01

**Soundness:** 3
**Presentation:** 3
**Contribution:** 3
**Rating:** 6
**Confidence:** 3

**Summary:**

This paper proposed a framework to scale up synthetic data generation for multi-hop question answering over table+text contexts.
It creates new tables by merging existing tables with atomic facts extracted from texts, then use a subset to verbalize as the text context.
Questions are generated using LLMs to build nested SQL queries bottom-up, then verbalize the queries that will yield result after execution. It also introduced an automated provenance-based approach to fix SQL query.

Contributions
1. The approach seems reasonable for constructing multi-hop type questions
2. Benchmark is challenging and shows there's still significant headroom for improvements.

**Strengths:**

1. Paper is well written and clear. Results demonstrate effectiveness of the dataset construction approach
2. Automated fixing and provenance-based fixing is a pretty creative idea
3. Substantial analysis of model failures on this dataset is also presented

**Weaknesses:**

1. seems to be missing the constructed dataset statistics? e.g. how many reference tables are constructed per source dataset?
2. scalability of this approach seems to be bottlenecked by number of source tables?

**Questions:**

Mostly minor suggestions on presentations, since this paper has a lot of details and analysis

1. What is the why-not based provenance approach (referenced paper only has why-provenance)? I can kind of infer from Figure 3 but maybe have some textual description would help?
2. How is provenance-based approach used to help fix the error => seems like the provenance analysis is added to LLM prompt to ask LLM to fix the query/relax the offending constraints, but it is unclear to me how this is used & what the inferior alternatives are doing when I first read through it -- especially in section 3.3.2 -- and Figure 3 only covers Provenance-based approach, so at first it was confusing how the alternatives are doing the fixes (plus figure 3 is small). Maybe adding some textual descriptions to clarify could help

---

> ### Author Response · Authors · 2025-11-19
> **Response to Reviewer kiAT**
>
> Dear Reviewer kiAT,
>
> Thank you for your valuable feedback. We have carefully addressed your suggestions and provided our detailed responses below. We believe these revisions, guided by your comments, will greatly improve the clarity and readability of our work. Please let us know if there is anything further we need to address to help you consider updating your score.
>
> ---
>
> **Response to W1:**
> Thank you for your comment. SPARTA constructs **9** reference tables for the NBA domain, **6** for Movie, and **5** for Medical;  We have added Table 1 in our revised manuscript, which reports the average number of rows per table across domains. To provide finer-grained dataset statistics, we have also added per-table row and column counts for all reference tables in Appendix D of the revised manuscript.
>
> ---
>
> **Response to W2:**
> To assess whether scalability is bottlenecked by the number of tables, we measured the average number of LLM calls needed to generate a single complete SPARTA query instance as the number of accessed tables increases. The cost grows by only +2.6 calls (from 1 to 2 tables), +3.9 (from 2 to 3), and +2.2 (from 3 to 4), indicating near-linear growth rather than an exponential blow-up. We have added this analysis to Appendix F of the revised manuscript.
>
> | # of Accessed Tables | Average # of LLM Calls per Query |
> |----------------------|----------------------------------|
> | **1**                | 2.3                          |
> | **2**                | 4.9 (+2.6)                   |
> | **3**                | 8.8 (+3.9)                   |
> | **4**                | 11.0 (+2.2)                  |
>
> ---
>
> **Response to Q1 & Q2:**
> In the database literature, why-not provenance is used to explain why certain expected tuples are missing from a query result by identifying the predicates or joins that block them [1, 2, 3]. While the cited paper focuses on why-provenance, we borrow the complementary notion of why-not provenance from this broader line of work. In SPARTA, we implement a lightweight variant that automatically infers the “expected” tuples from intermediate results, rather than requiring user-provided examples.
>
> As described in Section 3.3.2, our provenance-based refinement stage operates in three main steps:
> 1) Iteratively Relax Predicates: When a query returns an empty result, the provenance-based refinement module iteratively relaxes its predicates in reverse post-order until we obtain a subquery that produces a non-empty result.
> 2) Dynamically Sample Surviving Tuples: From this non-empty subquery, we dynamically sample one surviving tuple as the expected tuple.
> 3) Repair the Query Based on the Provenance Report: Finally, we perform a why-not provenance analysis to pinpoint the blocking predicate. This provenance report is then injected into the LLM prompt, and LLM is instructed to rewrite the problematic clause.
>
> By contrast, the inferior alternatives simply re-prompt the LLM after an execution failure or empty result without predicate-level feedback: they ask the LLM to “fix” the query globally, forcing it to guess which part is wrong. Our provenance-based approach, instead, provides a focused explanation of where the query fails, leading to more targeted and efficient repairs.
>
> [1] Bidoit, Nicole, Melanie Herschel, and Katerina Tzompanaki. "Query-based why-not provenance with nedexplain." Extending database technology (EDBT). 2014.
>
> [2] Chapman, Adriane, and H. V. Jagadish. "Why not?." Proceedings of the 2009 ACM SIGMOD International Conference on Management of data. 2009.
>
> [3] Lee, Seokki, et al. "A SQL-middleware unifying why and why-not provenance for first-order queries." 2017 IEEE 33rd International Conference on Data Engineering (ICDE). IEEE, 2017

---

> ### Author Response · Authors · 2025-11-27
> **A Gentle Reminder**
>
> Dear Reviewer kiAT,
>
> Thank you again for your thoughtful and constructive comments, which have been helpful for improving our submission.
>
> As the author–reviewer discussion period is coming to an end, we would like to kindly confirm that you have had a chance to review our detailed rebuttal. We believe that the main technical concerns you raised (missing constructed dataset statistics, concerns on scalability with increasing tables, insufficient explanation on provenance-based refinement) are now thoroughly addressed in our response and revisions.
>
> Additionally, because these updates and clarifications were made specifically in response to your constructive feedback, we kindly hope that they will encourage you to reconsider your evaluation. Should you have any remaining questions or reservations, please feel free to let us know at any time—we are fully committed to addressing all concerns to your satisfaction.
>
> Best,
>
> Authors

---

### Official Review · Reviewer_JYtH · 2025-11-01

**Soundness:** 4
**Presentation:** 4
**Contribution:** 4
**Rating:** 6
**Confidence:** 3

**Summary:**

This paper proposes SPARTA, a Text-Table Q&A dataset which is primarily generated by LLM prompts, with light human evaluation. This allows the authors to generate a large corpus of Q&A problems, over varying aspects of the problem instance. The authors provide extensive and high-quality evaluation of the dataset, including qualitative and ablation results.

**Strengths:**

Overall, this paper is very well written and I see great value in the proposed dataset.

1. The scope and scale of the dataset is seemingly quite novel. There is great care to the tested aspects of the dataset, and the overall size is sufficiently large for high-quality model comparison.

2. Each aspect of the dataset design is well motivated, and easy to interpret. This paper is interpretable for the general AI research reader.

3. The dataset evaluation is sufficient and provides several insightful, qualitative results.  e.g. Fig 5, Table 6.

4. The authors give a very extensive appendix with examples.

**Weaknesses:**

1. A primary weakness is the lack human-written and human-curated Q&A instances. While the synthetic generation methodology is impressive and well-evaluated, the authors could provide small curated data subsets for testing certain aspects of reasoning. For example, reasoning over ranges or negations, data inconsistencies (e.g. where text and tables have differing values for a specific fact), etc.

2. Similarly, While the scope of SPARTA is impressive and there is variation over structural, breadth, and clause diversity, more qualitative evaluation could be provided related to breadth in weakness (1). It is difficult to understand *what* model aspects are being tested (and not tested) over the entirety of the dataset.

3. The dataset doesn't include human-verified perturbation groups for robustness analysis. Since machine-guided perturbation models often change the result of complex queries, it would be valuable to validate over a subset of human-verified perturbations (e.g. query rephrasing that retains the same output).

4. Table 6 is somewhat concerning. Given poor text performance, is this problem primarily Tabular Q&A? Does SPARTA have a view where users can consistently use it for Tabular Q&A evaluation?(e.g. no comparability errors from users).

**Questions:**

1. Could you more succinctly summarize the aspects which are tested by this dataset, providing toy examples? (lengthy) examples are deep in the appendix. Can this be promoted to the main text in a better way than Table 1? (Perhaps the text around Table 1 can be more contrastive), or a taxonomic figure can be added.

2. Are there aspects which were omitted from the dataset at design-time?
a. robustness as described above
b. chain of thought reasoning
c. user evaluation to evaluate naturalness and scope of Q&A
d. grounded attribution and/or explanation quality
e. potential bias in evaluation, e.g. evaluation over generated sensitive attribute subgroups or subgroup-perturbations (curation scope r.e. Weakness 3 above)

Could you summarize the challenges or considerations in scoping the evaluation as you did?

**Details Of Ethics Concerns:**

The dataset doesn't include generated instances suitable for measuring bias in Text-Table Q&A. So downstream evaluations on SPARTA need an post-hock, and therefore incomparable, bias evaluation on their perturbation model etc of choice.

---

> ### Author Response · Authors · 2025-11-19
> **Response to Reviewer JYtH (1/2)**
>
> Dear Reviewer JYtH,
>
> Thank you for your constructive suggestions. We have carefully addressed each of your questions and performed additional analyses and experiments as recommended. Please find our detailed responses below. We believe these revisions, guided by your feedback, substantially enhance the clarity and empirical strength of our manuscript.
>
> ---
>
> **Response to W1:**
> Thank you for your valuable suggestion. We first note that SPARTA already provides comprehensive coverage of such reasoning patterns: our benchmark includes 2569 queries involving ranges and 242 involving negations. We further audited the SQL coverage and confirmed that all standard SQL operators (>, =, <>, <, >=, LIKE, NOT LIKE, <=, IN, NOT IN, EXISTS, NOT EXISTS) and aggregate functions (MAX, COUNT, AVG, MIN, SUM) are represented, except for “NOT LIKE” and “NOT EXISTS.”
>
> In response to your request for human-written, human-curated instances, we additionally constructed a small curated subset: we have added 30 human-generated queries for each of “NOT LIKE” and “NOT EXISTS,” as well as 30 human-generated queries specifically targeting range reasoning. These curated instances will be released together with the benchmark. On the other hand, queries requiring the detection of inconsistencies between text and tables are intentionally out of scope: we deliberately avoid duplicating the same facts across modalities to prevent redundant-modality issues, as discussed in Section 1.
>
> ---
>
> **Response to W2:** To make it clearer which reasoning aspects SPARTA tests, we have conducted targeted analyses on the range queries and negation phenomena highlighted in Weakness (1). For HProPro, the F1 scores on the negation subset and range subset are 28.7 and 32.9, respectively, versus an overall SPARTA (Oracle) average of 40.4, corresponding to drops of 28.3% and 18.6%. This quantitatively confirms that models are particularly weak on negation and range reasoning.
>
> | Category                                   | F1 (HProPro w/ GPT-5) |
> |--------------------------------------------|------------------------|
> | **Negation**(NOT LIKE, NOT EXISTS, NOT IN, <>) | 28.7 (-28.3%)   |
> | **Range**(>, <, >=, <=)                | 32.9 (-18.6%)   |
> | **SPARTA (Oracle)**                        | 40.4               |
>
> We also performed a qualitative case study by selecting representative queries for both negation and range and analyzing their failure patterns. These examples, together with a discussion of the specific capabilities they probe, have been added to Appendix H to clarify what kinds of reasoning SPARTA tests.
>
> ---
>
> **Response to W3:**
> To evaluate the model's robustness to linguistic variability, we have followed your recommendation and created a set of human-verified perturbations. We randomly sampled 100 queries from our benchmark and asked human annotators to rephrase each query while  preserving its core semantics and gold answer. We then evaluated the HProPro model with GPT-5 on both the original and the rephrased sets.
>
> | Category  | F1 (HProPro w/ GPT-5) |
> |-----------|-----------------------|
> | Original Questions  | 45.22         |
> | Rephrased Questions | 45.02 (-0.44%)|
>
> As shown in the table, the F1 score decreases from 45.22 on the original queries to 45.02 on the rephrased versions, a negligible drop of 0.44%. This suggests that, in SPARTA, performance is largely driven by the underlying table–text reasoning complexity rather than by sensitivity to surface-level phrasing. We have incorporated the details of this new experiment and its results into Appendix I of our revised manuscript.
>
> ---
>
> **Response to W4:** We would like to clarify that the "w/ Text Data" and "w/o Text Data" settings in our experiments denote query types, not different input data. Specifically, "w/ Text Data" refers to queries requiring cross-reasoning between tables and text to derive answers, while "w/o Text Data" refers to queries that can be resolved through table-only reasoning. Thus, the performance drop observed in Table 6 indicates limitations in the models' ability to integrate and reason across these two data modalities. To avoid confusion, we have renamed these settings to "Table–Text Cross Reasoning" and "Table-Only Reasoning," respectively, in the revised manuscript.
>
> ---
>
> **Response to Q1:** As requested, we have revised Figure 1 in the manuscript to summarize the key aspects evaluated by the SPARTA benchmark, namely, Tree-structured Reasoning, Large-scale Tables, and Analytical Operations, while including a representative toy example for each. Additionally, to make Table 1 more contrastive, we have extended it with a comparison of annotation error rates across benchmarks, showing that SPARTA, unlike existing datasets, exhibits no detected annotation errors.
>
> ---

---

> ### Author Response · Authors · 2025-11-19
> **Response to Reviewer JYtH (2/2)**
>
> **Response to Q2:**
>
> Our dataset considers the mentioned aspects as follows:
> - **Robustness as described above**: We have followed your recommendation in Weakness (3) and created a set of human-verified perturbations to evaluate the model's robustness to linguistic variability. We report robustness results on this subset in the revised manuscript.
> - **Chain of thought reasoning**: Queries often traverse multiple tables via joins and perform analytical operations, which encourages chain-of-thought–style reasoning in models, even though we do not supervise or evaluate intermediate natural-language chains explicitly.
> - **User evaluation to evaluate naturalness and scope of Q&A**: For the naturalness of SQL queries, we recruited three computer science graduate students with expertise in SQL and schema design. Each independently scored the same queries on a 1–5 scale, and we averaged their scores to obtain the final naturalness rating. For scope, we rely on the mechanically specified SQL query space: as noted in our response to Weakness (1), SPARTA covers all major SQL operators and aggregate functions. Because scope is defined by this formal query space, we did not conduct a separate user study on scope.
> - **Grounded attribution and explanation quality**: Since each natural language query corresponds to an SQL query, the process of deriving grounded answers from the dataset is guaranteed. Each natural-language question in SPARTA is paired with a ground-truth SQL query, which serves as an executable and fully interpretable reasoning trace over the underlying tables and text.
> - **Potential bias in evaluation**: Bias may exist in natural language variability, but this can be addressed as a separate benchmark issue. For instance, the Spider [1] Text-to-SQL benchmark has a variant, "Spider-Syn" [2], specifically designed to mitigate bias from synonym substitutions.
>
> [1] Yu, Tao, et al. "Spider: A Large-Scale Human-Labeled Dataset for Complex and Cross-Domain Semantic Parsing and Text-to-SQL Task." Proceedings of the 2018 Conference on Empirical Methods in Natural Language Processing. 2018.
>
> [2] Gan, Yujian, et al. "Towards Robustness of Text-to-SQL Models against Synonym Substitution." Proceedings of the 59th Annual Meeting of the Association for Computational Linguistics and the 11th International Joint Conference on Natural Language Processing (Volume 1: Long Papers). 2021.
>
> ---
>
> **Response to Q3:** Our main goal was to evaluate complex queries that require tree-structured multi-hop reasoning and advanced analytical operations over tables and text. To keep this realistic and automatically checkable at scale, we scoped SPARTA around what can be precisely expressed and executed in SQL.
>
> Concretely, as described in Section 1, we use nested SQL queries to represent multi-hop questions, covering four representative nesting patterns (Types N, A, J, and JA), and we ensure coverage of all basic SQL operators and aggregate functions (see our response to Weakness (1)).
>
> The main challenge was to generate such nested queries with rich structure and operator diversity while still guaranteeing realism and successful execution with valid results. This is why our framework includes two safeguards—provenance-based refinement and realistic-structure enforcement—which together allow us to efficiently produce semantically realistic, executable queries at scale.

---

> ### Author Response · Authors · 2025-11-27
> **A Gentle Reminder**
>
> Dear Reviewer JYtH,
>
> Thank you again for your thoughtful and constructive comments, which have been helpful for improving our submission.
>
> As the author–reviewer discussion period is coming to an end, we would like to kindly confirm that you have had a chance to review our detailed rebuttal. We believe that the main technical concerns you raised (absence of human-curated subsets for specific reasoning aspects, insufficient qualitative evaluation of negation and range reasoning, absence of robustness analysis against linguistic variability, concern over poor text performance in Table 6, suggestion for succinct summary of tested aspects, omitted aspects in dataset design, clarification of challenges in evaluation scoping) are now thoroughly addressed in our response and revisions.
>
> Additionally, because these updates and clarifications were made specifically in response to your constructive feedback, we kindly hope that they will encourage you to reconsider your evaluation. Should you have any remaining questions or reservations, please feel free to let us know at any time—we are fully committed to addressing all concerns to your satisfaction.
>
> Best,
>
> Authors

---

### Author Response · Authors · 2025-11-29
**Response to New Area Chair**

Dear New Area Chair,

First, we sincerely appreciate your time and effort in stepping in to handle this submission under these unexpected circumstances. We also deeply value the constructive feedback and insightful discussions with the reviewers, which have been instrumental in significantly strengthening our manuscript.

We are writing to summarize the status of our paper following the recent system reversion. Before the discussion period was halted, we had reached a consensus with all reviewers regarding the improved quality of our paper, securing updated or committed scores of **at least 6** from every reviewer.

Since the current system scores do not reflect the successful rebuttal, we provide a summary of the reviewer status at the time of the freeze.

1. **Status of Scores and Consensus**
- **Reviewer N2ch:** Explicitly updated the score to 6 on Nov 20, acknowledging that we had resolved the raised concerns.
- **Reviewer iHeT:** Explicitly committed to raising score to 6. On Nov 28, the reviewer stated s/he would raise the score from 4 to 6 once the edit option opened, noting we had addressed his/her concerns.
- **Reviewer JYtH & kiAT:** Positive (Score 6). Both were already positive. We thoroughly addressed their remaining concerns—including additional baselines and clarifications—to further strengthen the paper and solidify their support.

2. **Detailed Summary of Improvements & Resolved Concerns:** We addressed every concern raised. Below are the specific technical details and results added during the rebuttal:
- **Methodology Clarifications (Addressing N2ch, iHeT, kiAT)**
  - Clarification on "Low" Performance: We clarified that SPARTA is a benchmark framework, not a QA method. Thus, observed vulnerabilities (e.g., low F1 on Type-JA) are intended diagnostic findings revealing model gaps, not flaws in our method.
  - Why-Not Provenance (Novelty): We detailed our refinement strategy (Section 3.3.2), which adapts "why-not provenance" to identify specific predicates blocking results. This enables targeted repairs, significantly outperforming inefficient global re-prompting.
  - "Lightweight" vs. Rigor: We clarified that our validation is efficient, not less rigorous. Unlike HybridQA (where annotators read full tables/passages), SPARTA focuses on validating semantic alignment against pre-grounded SQL. This reduces annotation time to one-quarter of HybridQA, despite generating structurally more complex queries.
  - Atomic Fact Extraction: We clarified our dual strategy: using verified gold-standard tables (NBA) and deterministic templates (Movie/Finance) to ensure factuality without extraction errors.
- **Robustness & Experiments (Addressing N2ch, iHeT, JYtH)**
  - Generation Cost: We expanded evaluation to gpt-oss-20B, Llama-3.1-70B-Instruct, and gpt-oss-120B. Results confirm our approach reduces API calls by 18.8%–66.2% across all models (Appendix F).
  - Error Analysis: We conducted a fine-grained audit of end-to-end errors. The analysis identifies reasoning as the primary bottleneck (60.8%) compared to retrieval, with failure patterns consistent with the Oracle setting (Appendix G).
  - Analysis on Negation and Range Queries: We performed a targeted analysis on negation and range queries. Results show HProPro performs significantly worse on negation (28.7 F1) and range (32.9 F1) subsets compared to the average (40.4 F1). We added a qualitative case study of these failure patterns to Appendix H.
  - Linguistic Variability: We verified robustness against paraphrasing. The negligible F1 drop (45.22 $\to$ 45.02) confirms the model focuses on reasoning complexity rather than surface phrasing (Appendix I).
  - Table-Text Performance: We clarified that performance drops in "Table-Text" settings reflect the model’s limitation in cross-modal integration, not an inability to process text.
- **Benchmark Design & Statistics (Addressing N2ch, iHeT, kiAT, JYtH)**
  - Scalability: We analyzed cost growth as the number of tables increased (1 to 4 tables). The cost grows nearly linearly (+2.2 to +3.9 calls), disproving concerns about exponential complexity.
  - Logical Bridging: We demonstrated that performance drops are uniform across domains, highlighting systemic model challenges. Logical consistency is ensured via explicit foreign keys and established join relationships.
  - New Human-Curated Subset: To ensure complete operator coverage, we augmented the benchmark with 90 human-curated queries specifically targeting NOT LIKE, NOT EXISTS, and ranges.
  - Dataset Expansion: To facilitate broader adoption, we committed to making the entire data generation framework and code publicly available.
3. **Conclusion:** Through the rebuttal process, we achieved a consistent consensus of acceptance across all reviewers. We are confident that SPARTA will lay a critical foundation for active and rigorous future research in this domain.

We hope this summary assists you in your final decision-making.

---

### Public Comment · ~Cyprien_Louis_René_Fourcroy1 · 2026-05-20
**Concerned about evaluation of models**

Overall, this is a well-written and interesting paper. However, I have some concerns regarding the evaluation setup for certain table-text QA models, particularly Odyssey.

From the released code, Odyssey appears to use `nba/dataset/text_data.json` as its textual context. This file contains game summaries for 7,202 NBA games, but these summaries are far less detailed than the grounding tables `nba_player_game_stats`, `nba_team_game_stats`, and `nba_game_information`, which contain fine-grained structured statistics for each game and player.

Moreover, some information described in the passages (for example, fouls or narrative events mentioned in the summaries) does not seem to appear in the grounding tables. Therefore, the textual passages (C_1, ..., C_P) and the grounding tables (GT) do not appear to be fully equivalent representations of the same information.

From the code, it also seems that Odyssey excludes grounding tables from direct table serialization and instead mainly reasons over passages plus source tables. If this interpretation is correct, the evaluation setting differs from a setup where models can directly access the grounding tables relationally.

It would be very helpful if the paper could clarify:

1. whether the grounding tables are intended to be complete relational equivalents of the passages,
2. how the NBA grounding tables were actually constructed,
3. and precisely which components (source tables, grounding tables, passages) are exposed to Odyssey during inference.

---

### Meta-Review · Area_Chair_K1fJ · 2026-01-06

**Summary:**

The paper presents a large-scale (thousands of instances) dataset for multi-hop table QA. The construction process relies on augmenting tables with facts extracted from accompanying text, using an LLM pipeline to synthesize multi-hop SQL queries and verbalize these queries as natural language questions. Human validation is comparatively lightweight. The paper compares performance of state-of-the-art models on the proposed dataset and on HybridQA and OTT-QA, showing that performance is substantially lower on the proposed dataset.

Strengths:
Multiple reviewers appreciated the large-scale nature of the dataset and felt that the creation process was well-motivated and creative. After discussion, reviewers also agreed that the quality of the individual steps of the construction process was high. Reviewers also felt that the analysis was comprehensive, and the results and failure analyses interesting.

**Reviewer Concerns:**

A main concern before the response was the reliance on LLMs, which could create unnatural examples, and be inefficient. However, the response addressed these: pointing to the user evaluation of naturalness, and showing both that the number of LLM calls scales roughly linearly with the number of tables involved in a question and the approach's "provenance-based refinement" reduces the number of required LLM calls.

**Reviewer Scores:**

The two reviewers that gave 4s did indicate that they would raise their scores. So I expect that the final scores would be 6 / 6 / 6 / 6.

---

### Decision · Program_Chairs · 2026-01-26

Accept (Poster)